# JanusDNA: A Powerful Bi-directional Hybrid DNA Foundation Model

**Qihao Duan**[1,4,5], **Bingding Huang**[2], **Zhenqiao Song**[3],
**Irina Lehmann**[1], **Lei Gu**[4][†], **Roland Eils**[1,5,6,7][†], **Benjamin Wild**[1][†]
[1]Berlin Institute of Health, Charité – Universitätsmedizin Berlin
[2]College of Big Data and Internet, Shenzhen Technology University
[3]Language Technologies Institute, Carnegie Mellon University
[4]Epigenetics Laboratory, Max Planck Institute for Heart and Lung Research
[5]Department of Mathematics and Computer Science, Freie Universität Berlin
[6]Health Data Science Unit, Heidelberg University Hospital and BioQuant
[7]Intelligent Medicine Institute, Fudan University
[†]Corresponding authors
{qihao.duan, irina.lehmann, roland.eils, benjamin.wild}@bih-charite.de
huangbingding@sztu.edu.cn zhenqias@andrew.cmu.edu
lei.gu@mpi-bn.mpg.de

## Abstract

Large language models (LLMs) have revolutionized natural language processing
and are increasingly applied to other sequential data types, including genetic se-
quences. However, adapting LLMs to genetics presents significant challenges.
Capturing complex genomic interactions requires modeling long-range global de-
pendencies within DNA sequences, where interactions often span over 10,000 base
pairs, even within a single gene. This poses substantial computational demands
under conventional model architectures and training paradigms. Additionally,
traditional LLM training approaches are suboptimal for DNA sequences: autore-
gressive training, while efficient for training, only supports unidirectional sequence
understanding. However, DNA is inherently bidirectional. For instance, bidirec-
tional promoters regulate gene expression in both directions and govern approxi-
mately 11% of human gene expression. Masked language models (MLMs) enable
bidirectional understanding. However, they are inefficient since only masked
tokens contribute to loss calculations at each training step. To address these
limitations, we introduce **JanusDNA, the first bidirectional DNA foundation
model built upon a novel pretraining paradigm**, integrating the optimization effi-
ciency of autoregressive modeling with the bidirectional comprehension capability
of masked modeling. JanusDNA's architecture leverages **a Mamba-Attention
Mixture-of-Experts (MoE) design**, combining the global, high-resolution context
awareness of attention mechanisms with the efficient sequential representation
learning capabilities of Mamba. The MoE layers further enhance the model's
capacity through sparse parameter scaling, while maintaining manageable com-
putational costs. Notably, **JanusDNA can process up to 1 million base pairs
at single-nucleotide resolution on a single 80GB GPU using its hybrid ar-
chitecture**. Extensive experiments and ablation studies demonstrate that Janus-
DNA achieves new state-of-the-art performance on three genomic representation
benchmarks. Remarkably, JanusDNA surpasses models with **250x** more acti-
vated parameters, underscoring its efficiency and effectiveness. Code available at
https://github.com/Qihao-Duan/JanusDNA.

39th Conference on Neural Information Processing Systems (NeurIPS 2025).

# 1  Introduction

Modeling the *language* of DNA is crucial for investigating its biological function, offering potential advancements in understanding genotype-phenotype associations, disease diagnosis, and new drug development [1]. Large language models (LLMs) have demonstrated remarkable success in large-scale nonlinear representation learning for natural language processing (NLP) tasks, such as text generation, translation, and summarization [2, 3, 4]. This success has inspired researchers to explore LLM applications in other domains [5, 6], including bioinformatics [7]. However, applying general LLMs directly to DNA sequence data is challenging. DNA sequences are represented as series of nucleotides, lacking clear semantic meaning and posing challenges for interpretation by LLMs. Additionally, non-coding regions in DNA can be remotely related to coding regions both upstream and downstream, necessitating models capable of processing long-range dependencies while maintaining bidirectional understanding. These factors make it challenging to apply LLMs directly to DNA sequence data without significant modifications or adaptations [8]. Some models have been developed specifically for DNA sequence representation [9, 10, 11]. However, these models still face some limitations.

**Current limitations**   (1) **Limited Sequence Length and Low Resolution:** Capturing complex genomic interactions requires modeling long-range dependencies within DNA sequences, where interactions can span over 10,000 base pairs even within a single gene [12]. This necessitates a model that can effectively handle long-range dependencies and relationships within the sequence. However, many current models solely rely on global attention mechanisms inspired by their superior success in natural language applications [13, 14], yet they often struggle to effectively process long genomic sequences and uncover meaningful long-range interactions [11]. K-mer tokenization is frequently used to expand the context window of DNA sequence models [9, 10]. However, this method introduces a trade-off between sequence length and resolution [15], potentially leading to the loss of crucial information, especially in cases where single nucleotide polymorphisms (SNPs) are essential for understanding gene function. (2) **Unidirectional Understanding:** Many genomic processes are influenced by bidirectional interactions, with essential regulatory elements located both upstream and downstream of key genomic regions. For example, bidirectional promoters initiate transcription in both orientations [16, 17]. Additionally, intergenic enhancers transcribed predominantly bidirectionally often function as weak promoters in both directions. Conversely, for elements with unidirectional transcription (both enhancers and promoters), transcription direction typically correlates with the orientation in which the element functions as a promoter in vivo [18]. Accurately modeling these bidirectional interactions is crucial for understanding genomic function and making precise predictions, imposing significant demands on model architecture and training strategies [8]. However, some existing decoder-only models based on State Space Models (SSMs) [15, 19, 20] are primarily unidirectional or limited in their capacity to effectively understand bidirectional context, thus constraining their ability to comprehensively capture these complex interactions. (3) **Low Training Efficiency:** Long-range modeling and bidirectional understanding of DNA sequences both require substantial computational resources and memory, especially for those requiring attention for a more global representation. Therefore, training efficiency significantly impacts the model's performance, especially under limited computational resources. Most bidirectional models are trained using masked training paradigms, such as BERT [21], which utilize only a small fraction of tokens (typically 15%) for loss calculation at each step. This limitation can hinder the model's ability to learn effectively from the entire sequence within limited training steps, thereby requiring more training epochs to adequately cover the full training data.

Additionally, the masking process itself introduces extra computational overhead. In contrast, autoregressive (next-token prediction) training is more efficient, as nearly all tokens contribute to the loss at each training step, allowing the model to learn more effectively within a fixed number of steps as sequence length increases [22]. However, it's important to note that autoregressive models are inherently unidirectional, limiting their ability to incorporate bidirectional context.

In response to the aforementioned issues, we introduce **JanusDNA**[1], the first bidirectional DNA foundation model built upon a novel pretraining paradigm. Our architecture employs two principal strategies: (1) **Hybrid Architecture:** To achieve powerful global understanding while maintaining

---

[1]Janus, the ancient Roman god of transitions and duality, is symbolized by two faces gazing in opposite directions.

computational efficiency for long contexts, we integrate the strengths of state-space models (SSMs) [23] and Mixture-of-Experts (MoE) designs [24, 25] into attention mechanisms [26], enabling the model to effectively capture long-range global dependencies and complex interactions within DNA sequences. (2) **Bidirectional Efficient Training:** While preserving the bidirectional understanding of DNA sequences typically achieved through masked training, we significantly improve learning efficiency by computing the loss over all tokens in each training, same as in autoregressive modeling. Notably, **JanusDNA is capable of processing up to 1 million base pairs at single-nucleotide resolution with global attention on a single 80GB GPU**, making it suitable for large-scale understanding in genomic research. We evaluated JanusDNA on 35 diverse genomic tasks to showcase its superior global understanding as well as long-range representation ability.

In summary, our contributions are as follows:

- We propose **JanusDNA**, a novel bidirectional DNA foundation model capable of capturing global long-range dependencies and interactions at single-nucleotide resolution.

- We introduce an efficient **Hybrid Mamba-Attention-MoE architecture** designed for processing ultra-long genomic sequences within practical computational budgets.

- We present **Janus Modeling**, a novel and efficient pretraining paradigm that effectively combines the strengths of autoregressive and masked modeling, facilitating effective global bidirectional learning.

- We demonstrate **state-of-the-art performance** across diverse genomic benchmarks, outperforming significantly larger models. In particular, JanusDNA **significantly surpasses the expert model Enformer on eQTL prediction tasks**, despite having far fewer parameters.

## 2 Preliminary and Related Work

### 2.1 Large Language Model Pretraining Paradigms

**Autoregressive Language Modeling (ALM)** is a generative pretraining paradigm in which the model predicts the next token in a sequence given all previous tokens. Trained on large corpora, the model learns statistical properties to generate coherent text. The training objective is:

$$\mathcal{L}_{\text{CLM}} = -\sum_{t=1}^{T} \log P(x_t | x_1, x_2, \ldots, x_{t-1}), \tag{1}$$

where $T$ is the sequence length, and $x_t$ denotes the token at position $t$. The model generates text by sampling from the learned probability distribution over the vocabulary at each time step [15, 19, 20, 4, 27]. Each token contributes to the overall loss, and the model minimizes the average loss across all tokens. However, to maximize generative performance [28], ALM is unidirectional, causing a limited ability to model bidirectional contexts, which is crucial for DNA sequence understanding [1, 29].

**Masked Language Modeling (MLM)** is a non-causal pretraining paradigm where the model predicts masked tokens in a sequence using surrounding context. The training objective is:

$$\mathcal{L}_{\text{MLM}} = -\sum_{i=1}^{N} \log P(x_i | x_{j_1}, x_{j_2}, \ldots, x_{j_k}), \tag{2}$$

where $N$ is the total number of tokens, $x_i$ is the masked token, and $x_{j_1}, x_{j_2}, \ldots, x_{j_k}$ are the unmasked tokens. This approach enables the model to learn bidirectional representations, capturing dependencies in both directions [1, 28, 29, 9, 10, 30, 11, 31]. However, MLMs mostly follow the BERT-style training paradigm, which masks a fixed percentage of tokens in the input sequence, e.g., 15% [8, 9, 10]. This can lead to inefficiencies, as only a small fraction of the data is used for loss computation during each iteration. In contrast, autoregressive training paradigms take advantage of nearly the entire data, significantly improving training efficiency and overall performance.

**A detailed review of related work on DNA language models** is provided in Appendix A.

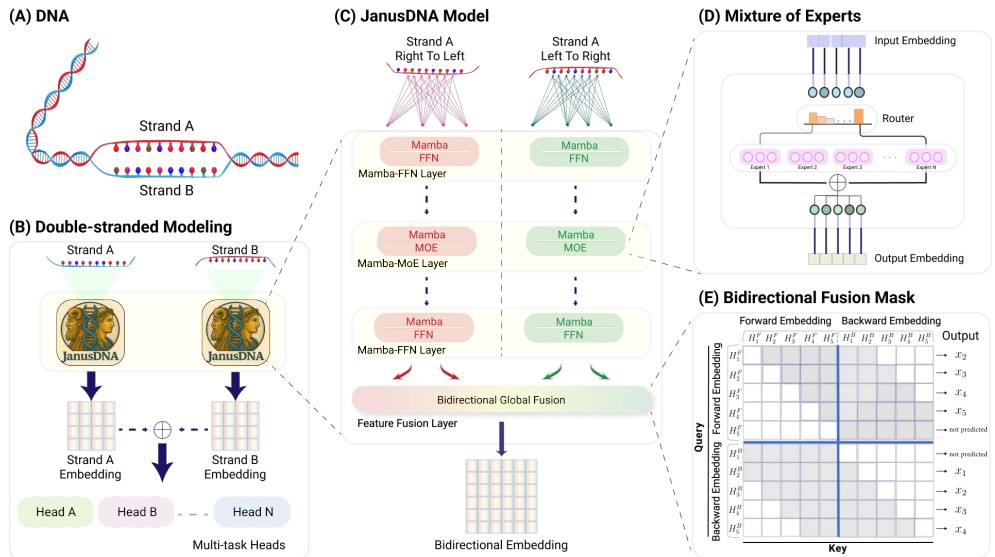

Figure 1: **The JanusDNA Architecture for Bidirectional DNA Modeling.** JanusDNA employs a hierarchical bidirectional strategy to comprehensively model DNA sequences. **(A)** DNA, with its inherent double-stranded nature. **(B)** The model processes both the forward and reverse complement strands independently in parallel to capture complete biological context, with their embeddings subsequently combined for downstream tasks. **(C)** The core JanusDNA model architecture processes a single input strand using parallel left-to-right and right-to-left pathways. Each pathway consists of Mamba-FFN and Mamba-MoE layers for effective and efficient sequential encoding. **(D)** The MoE architecture enhances model capacity and specialization by dynamically and sparsely routing inputs to a subset of expert networks, enabling efficient computation and improved representation learning. **(E)** The Bidirectional Global Fusion mechanism, utilizing a specific attention mask, integrates the forward and backward representations from (C) to ensure that each nucleotide's embedding is informed by its complete sequence context.

## 3   JanusDNA

We propose Janus modeling, an efficient bidirectional training method with global attention, and JanusDNA, a powerful hybrid DNA foundation model.

As illustrated in Figure 1, JanusDNA processes bidirectional DNA sequences from both left-to-right and right-to-left using two independent stacks of Mamba and Mixture-of-Experts (MoE) layers. These stacks generate forward and backward representations independently, ensuring no information leakage. The two representations are then fused to create a unified representation that encapsulates bidirectional information. Each token position is predicted based on all upstream and downstream tokens. The following sections detail the efficient bidirectional training method with global attention – **Janus Modeling**, and the hybrid Mixture-of-Experts (MoE) architecture – **JanusDNA**.

### 3.1   Bidirectional Efficient Training

As discussed in Section 2, conventional pretraining paradigms face a trade-off: Masked Language Models (MLMs) offer bidirectional understanding but suffer from low training efficiency due to sparse loss signals, especially for those requiring global attention, while Autoregressive Models are efficient in training but inherently unidirectional. To overcome this, we introduce *Janus modeling*, a novel pretraining objective designed to achieve efficient, fully bidirectional sequence understanding with global attention, as illustrated in Figure 2.

The core idea of Janus modeling is to predict *every* token $x_t$ in a sequence of length $T$ using its complete bidirectional context, i.e., all tokens preceding $x_t$ ($x_1, \ldots, x_{t-1}$) and all tokens succeeding $x_t$ ($x_{t+1}, \ldots, x_T$). The training objective is therefore:

$$\mathcal{L}_{\text{bidirectional}} = -\sum_{t=1}^{T} \log P(x_t | x_1, \ldots, x_{t-1}, x_{t+1}, \ldots, x_T) \tag{3}$$

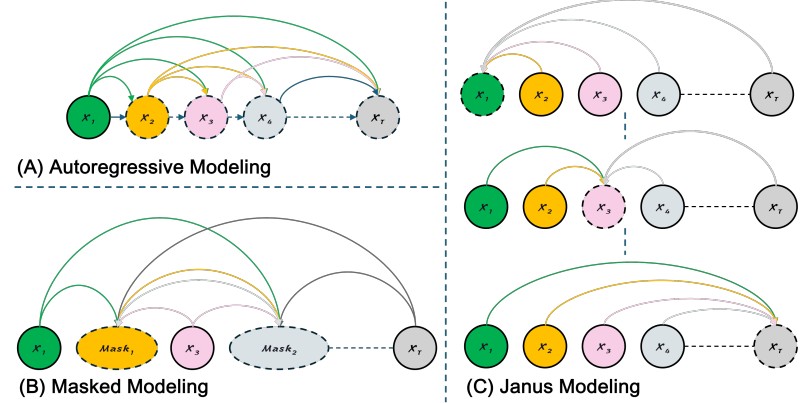

**(A) Autoregressive Modeling**

**(B) Masked Modeling**

**(C) Janus Modeling**

Figure 2: **Modeling Interpretation.** Janus modeling treats each token as a target for loss calculation, enabling higher training efficiency compared to masked modeling by full sequence learning while keeping bidirectional context understanding.

This objective ensures that every token contributes to the loss, maximizing training efficiency, while simultaneously demanding bidirectional comprehension.

To realize this objective, Janus modeling employs a two-stage process: independent context encoding followed by a global fusion mechanism:

**Independent Context Encoding:**  The input sequence $X = (x_1, \ldots, x_T)$ is processed by two parallel and independent stacks of layers (detailed in Section 3.2):

- A **forward pass** processes the sequence from left-to-right, generating a sequence of hidden states $\mathcal{R}_{\text{fwd}} = (H_1^F, H_2^F, \ldots, H_T^F)$. Each hidden state $H_t^F$ is a function of the input tokens $(x_1, \ldots, x_t)$ and primarily captures information about the left context of $x_t$.

$$H_t^F = \text{ForwardEncoder}(x_1, \ldots, x_t) \tag{4}$$

- A **backward pass** processes the sequence from right-to-left, generating a sequence of hidden states $\mathcal{R}_{\text{bwd}} = (H_1^B, H_2^B, \ldots, H_T^B)$. Each hidden state $H_t^B$ is a function of the input tokens $(x_T, \ldots, x_t)$ and primarily captures information about the right context of $x_t$.

$$H_t^B = \text{BackwardEncoder}(x_T, \ldots, x_t) \tag{5}$$

These two sets of representations, $\mathcal{R}_{\text{fwd}}$ and $\mathcal{R}_{\text{bwd}}$, are generated independently, ensuring no premature information leakage between past and future contexts before the explicit fusion step. The entire model is trained end-to-end using $\mathcal{L}_{\text{bidirectional}}$ from Equation 3.

**Bidirectional Global Fusion:**  To compute $P(x_t | x_1, \ldots, x_{t-1}, x_{t+1}, \ldots, x_T)$ for each $x_t$ as per Equation 3, the left-context information captured in $\mathcal{R}_{\text{fwd}}$ and the right-context information captured in $\mathcal{R}_{\text{bwd}}$ must be integrated. This is organically achieved via a global attention mechanism, specifically implemented with FlexAttention [26] for efficiency. The representations from both passes, $\mathcal{R}_{\text{fwd}} = (H_1^F, \ldots, H_T^F)$ and $\mathcal{R}_{\text{bwd}} = (H_1^B, \ldots, H_T^B)$, are concatenated to form a combined input sequence for the attention layer: $R_{\text{input}} = [H_1^F, \ldots, H_T^F, H_1^B, \ldots, H_T^B]$. This $R_{\text{input}}$ sequence has a length of $2T$. The core of the fusion lies in a carefully designed attention mask, $\mathcal{M}_{ij}$, which dictates how tokens in $R_{\text{input}}$ can attend to each other. This mask ensures that the prediction for $x_t$ is based only on $H_k^F$ for $k < t$ and $H_j^B$ for $j > t$, preventing information leakage. The mask, also illustrated in Figure 1(E), is defined as:

$$\mathcal{M}_{ij} = \begin{cases} q_{\text{idx}} \geq kv_{\text{idx}}, & \text{if } kv_{\text{idx}} < T \text{ and } q_{\text{idx}} < T, \\ q_{\text{idx}} \leq kv_{\text{idx}}, & \text{if } kv_{\text{idx}} \geq T \text{ and } q_{\text{idx}} \geq T, \\ kv_{\text{idx}} \geq T + q_{\text{idx}} + 2, & \text{if } kv_{\text{idx}} \geq T \text{ and } q_{\text{idx}} < T, \\ q_{\text{idx}} \geq kv_{\text{idx}} + T + 2, & \text{if } kv_{\text{idx}} < T \text{ and } q_{\text{idx}} \geq T. \end{cases} \tag{6}$$

Here, $q_{\text{idx}}$ and $kv_{\text{idx}}$ are the 0-indexed indices of the query and key-value pairs within the $2T$-length $R_{\text{input}}$, respectively. $T$ is the original sequence length. The mask $\mathcal{M}_{ij}$ is a binary matrix where allowed

attentions are 1 and disallowed are 0 (or $-\infty$ after softmax). The first two cases handle causal attention within the $\mathcal{R}_{\mathrm{fwd}}$ and $\mathcal{R}_{\mathrm{bwd}}$ segments, respectively. The third and fourth cases manage the cross-attention between $\mathcal{R}_{\mathrm{fwd}}$ and $\mathcal{R}_{\mathrm{bwd}}$ segments, precisely controlling information flow to maintain the integrity of bidirectional prediction without information leakage relative to the token being predicted.

The output of this attention mechanism provides a fused representation $H_t^{\mathrm{final}}$ for each token $x_t$, which is then used to make the final prediction following a repositioning step, where the representations of the same token are summed, except for the first and last tokens, due to their representations containing information from only one direction, while optimizing $\mathcal{L}_{\mathrm{bidirectional}}$.

This Janus modeling approach, as conceptually depicted in Figure 2 (C), enables each token to be a learning target informed by its full bidirectional context, thereby enhancing training efficiency compared to traditional MLMs (Figure 2 (B)) and overcoming the limitations of unidirectional models (Figure 2 (A)).

**Empirical Validation of Training Efficiency** To empirically assess the learning efficiency of Janus modeling against conventional bidirectional approaches, masked modeling, we conducted a comparative experiment focused on last-token prediction. This task was chosen as it allows a direct comparison: both a standard

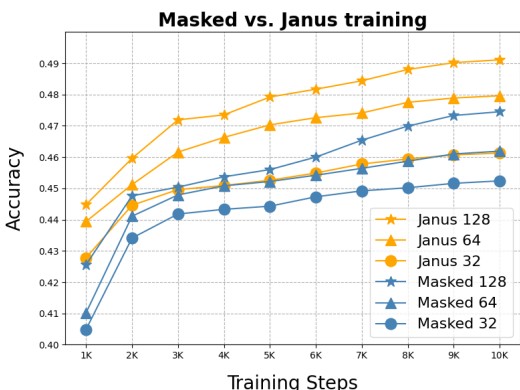

Figure 3: **Superior Learning Efficiency of Janus Modeling.** Comparison of last-token prediction accuracy between Janus modeling and conventional masked modeling over 10k training steps. Janus modeling consistently achieves higher accuracy for the same model architecture and training duration, demonstrating its enhanced efficiency in learning from sequence data. The number in the legend indicate hidden dimention.

masked language model and our Janus modeling approach can predict the final token $x_T$ given the preceding context $x_1, \ldots, x_{T-1}$, ensuring a fair basis for evaluation.

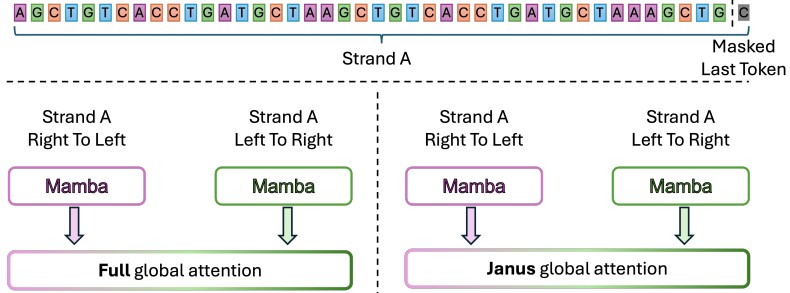

Figure 4: **Janus and Masked Modeling Efficiency Validation.** Both models are pre-trained from scratch using identical hyperparameter settings, with the only difference being the masking strategy applied in the final fusion attention layer. Last-token prediction is used to enable a fair comparison of learning efficiency between the two models.

For this demonstration, we configured two model variants as Figure 4: **Janus Model**: A single-layer bidirectional Mamba architecture equipped with a FlexAttention layer [26] for bidirectional fusion, utilizing the Janus-specific attention mask $\mathcal{M}_{ij}$ (Equation 6). This model predicts $x_t$ based on the fused bidirectional context of $x_1, \ldots, x_{t-1}, x_{t+1}, \ldots, x_T$ as per the Janus modeling objective. **Masked Language Model**: As a baseline, we construct a comparable single-layer Mamba architecture followed by the same FlexAttention layer without an attention mask, trained using the conventional masked language modeling (MLM) objective, where a fraction of tokens (typically 15%) contribute to the loss.

Both models were trained on the human reference genome (HG38) [32] for 10,000 steps. Each training sample had a context length of 131,072 tokens, processed with a batch size of 1. We evaluated performance across three hidden dimensions: 32, 64, and 128, keeping other hyperparameters

consistent. Pre-training for Janus models, benefited from their sparse attention masks, takes around 27 minutes per 1,000 steps, nearly twice as fast as masked models.

For evaluation, both the masked and Janus models are given the full input sequence with the last token $x_T$ masked. This evaluation is conducted on a test set containing 1,920 sequences.

The results, presented in Figure 3, clearly demonstrate that models trained with the Janus modeling method significantly outperformed those trained with the masked modeling approach in prediction accuracy, given the same number of training steps. This finding substantiates that Janus modeling is more effective at leveraging the DNA sequences for learning, thereby achieving superior training efficiency while maintaining robust bidirectional understanding.

### 3.2 Hybrid Mixture-of-Experts (MoE) Model

After introducing the bidirectional efficient training method, we developed a bidirectional backbone model to enhance sequence representation. Leveraging the efficient bidirectional fusion method, we propose JanusDNA, a hybrid model that integrates the strengths of Mamba, Attention, and MoE.

**Architecture** As illustrated in Fig. 1, JanusDNA incorporates three primary components for sequence representation: Mamba, MoE, and Attention. Mamba, a State Space Model (SSM) [23], efficiently encodes input sequences using high-dimensional parameters. Compared to traditional Transformer architectures, Mamba is more memory- and computationally efficient, making it particularly suitable for processing long DNA sequences.

The Mixture-of-Experts (MoE) architecture provides a scalable approach to significantly increase model capacity without proportionally increasing computational costs during training and inference [33]. In JanusDNA, MoE layers replace the feedforward network (FFN) layers in the Mamba blocks at a specific ratio following [24], achieving a balance between performance and efficiency. To ensure balanced utilization of the experts, an auxiliary loss is introduced [34], encouraging the model to distribute input data evenly across all experts. This auxiliary loss is computed based on the router logits, which represent the probabilities of selecting each expert for a given input, as shown in Eq. 7. Given $N$ experts indexed by $i = 1$ to $N$ and a batch $\mathcal{B}$ with $T$ tokens, the auxiliary loss is computed as the scaled dot-product between vectors $f$ and $P$:

$$\mathcal{L}_{\text{total}} = \alpha \cdot N \cdot \sum_{i=1}^{N} f_i \cdot P_i \tag{7}$$

where $f_i$ is the fraction of tokens dispatched to expert $i$,

$$f_i = \frac{1}{T} \sum_{x \in \mathcal{B}} \mathbb{1}\{\arg\max p(x) = i\}, \tag{8}$$

and $P_i$ is the fraction of the router probability allocated to expert $i$,

$$P_i = \frac{1}{T} \sum_{x \in \mathcal{B}} p_i(x). \tag{9}$$

Here, $p_i(x)$ represents the probability of routing token $x$ to expert $i$, while $T$ is the total number of tokens in the batch $\mathcal{B}$. This auxiliary loss encourages balanced utilization of all experts, improving overall model performance.

Bidirectional sequences are processed through independent stacks of Mamba and MoE layers, each designed to enhance the model's representational capacity. The Mamba layers efficiently capture local contextual dependencies within the sequence, leveraging their memory-efficient state-space modeling, while the MoE layers provide sparse scaling to enhance the model's representational capacity without incurring proportional computational overhead.

The forward and backward representations generated by these layers are fused using FlexAttention [26], an optimized attention mechanism that supports sparse masks with reduced memory consumption. This fusion enables the model to integrate both local and long-range global information streams, resulting in a comprehensive bidirectional representation for improved performance.

**Reverse Complement** In the double-helix DNA structure, each strand contains semantically equivalent information, with the *reverse complement* (RC) of a strand oriented in the opposite direction and its bases complemented relative to the *forward* strand (A paired with T, and C paired with G) [8]. However, recognizing both the forward and RC versions of non-palindromic motifs, such as *GATA* and *TATC*, poses a significant challenge, as it is akin to learning two distinct motifs [35]. To address this, we adopt a post-hoc reverse complement representation strategy [36]. Specifically, the DNA sequence and its reverse complement are processed in parallel using the identical model. The resulting representation vectors are then pooled to form a unified, enriched representation as shown in Figure 1 (B). This approach enables the model to effectively learn from both the original and reverse complement sequences, improving its ability to capture intricate patterns and relationships within DNA sequences, thereby enhancing performance across various tasks. We further conduct ablation experiments on reverse complement design in Appendix B.1.3.

## 4 Experiments

### 4.1 Pre-training on Human Reference Genome

To ensure a fair comparison with prior work, we pre-train our model on only the human reference genome (HG38 [32]) following the training setup described in [8]. Specifically, we adopt single nucleotide-level tokenization to capture high-resolution input sequences and avoid overlooking critical DNA information that may be lost when using k-mer tokenization, commonly used in attention-based models [15]. Additionally, single nucleotide-level tokenization is employed to facilitate downstream research on Single Nucleotide Polymorphisms (SNPs). Please note that performance on downstream tasks depends on the model architecture, as well as the composition and diversity of the pretraining data [11, 19, 20, 10]. Here, we specifically focus only on the architecture, and thus use only the human reference genome for pretraining to ensure a fair comparison.

### 4.2 Downstream Tasks

We evaluate our model on three different benchmarks: Genomic Benchmark [37], Nucleotide Transformer Benchmark [11], and DNALONGBENCH [38]. We follow all benchmark settings of Genomic Benchmark and Nucleotide Transformer Benchmark as described in [8]. Accordingly, we adopt the reported results from [8] as our reference. Considering the practical computational cost of sparse MoEs, we adjust the model's hidden size to match or slightly reduce the number of activated parameters compared to the baseline [8], ensuring a fair comparison. As we introduce the model with a middle attention layer in the ablation experiments (Section B.1.1), we present results for models both with and without mid-attention on the Genomic Benchmark and Nucleotide Transformer Benchmark.

#### 4.2.1 Genomic Benchmark

Table 1: Genomic Benchmarks. Top-1 accuracy (↑) across 5-fold cross-validation (CV) for a supervised CNN baseline, pretrained HyenaDNA, Caduceus models, ConvNova and JanusDNA models. Best values per task are **bolded**, second best are underlined. Error bars indicate the difference between the maximum and minimum values across 5 random seeds used for CV.

| MODELS ACTIVATED PARAM | CNN (264K) | HYENADNA (436K) | CADUCEUS PH (470K) | CADUCEUS PS (470K) | CONVNOVA (386K) | JANUSDNA MLP W/ MID-ATTN (426K) | JANUSDNA MLP W/O MID-ATTN (431K) |
|---|---|---|---|---|---|---|---|
| MOUSE ENHANCERS | 0.715±0.087 | 0.780±0.025 | 0.754±0.074 | **0.793**±0.058 | 0.784±0.009 | 0.770±0.048 | 0.769±0.029 |
| CODING VS. INTERGENOMIC | 0.892±0.008 | 0.904±0.005 | 0.915±0.003 | 0.910±0.003 | **0.943**±0.001 | 0.912±0.003 | 0.911±0.001 |
| HUMAN VS. WORM | 0.942±0.002 | 0.964±0.002 | **0.973**±0.001 | 0.968±0.002 | 0.967±0.002 | 0.971±0.001 | 0.971±0.001 |
| HUMAN ENHANCERS COHN | 0.702±0.021 | 0.729±0.014 | **0.747**±0.004 | 0.745±0.007 | 0.743±0.005 | 0.741±0.005 | 0.742±0.006 |
| HUMAN ENHANCER ENSEMBL | 0.744±0.122 | 0.849±0.006 | 0.893±0.008 | **0.900**±0.006 | **0.900**±0.004 | 0.897±0.004 | 0.899±0.004 |
| HUMAN REGULATORY | 0.872±0.005 | 0.869±0.012 | 0.872±0.011 | 0.873±0.007 | 0.873±0.002 | **0.877**±0.005 | 0.868±0.008 |
| HUMAN OCR ENSEMBL | 0.698±0.013 | 0.783±0.007 | **0.828**±0.006 | 0.818±0.006 | 0.793±0.004 | 0.822±0.003 | 0.824±0.001 |
| HUMAN NONTATA PROMOTERS | 0.861±0.009 | 0.944±0.002 | 0.946±0.007 | 0.945±0.010 | 0.951±0.003 | **0.957**±0.004 | 0.954±0.010 |

We start with the Genomic Benchmark, which is a collection of 8 regulatory element classification tasks with sequence lengths mostly ranging from 200 to 500, and one up to 4,776. We take the hidden

state embedding of the final layer and apply a pooling layer on sequences to obtain a fixed-length representation. We then apply a linear layer to map the representation to the number of classes for each task. We perform 5-fold cross-validation for each task using the same seed as [8]. As shown in Table 1, our model achieves state-of-the-art performance on 3 out of 8 tasks, outperforming the previous best model, while the remaining tasks are close to the best model. Please note that this benchmark is already quite saturated, as shown in Table 1, and we do not expect improvements in pretraining to meaningfully improve benchmark performance further.

Table 2: Nucleotide Transformer Tasks. Performance (↑) across 10-fold CV for Enformer, DNABERT-2, Nucleotide Transformer v2, HyenaDNA, Caduceus-PH, ConvNova, and JanusDNA$_{mlp}$. Metrics vary by task: MCC for histone markers and enhancer annotation, F1-score for promoter annotation and splice site acceptor/donor, and accuracy for splice site "all". Best values per task are **bolded**, second best are *italicized*. Given the disparity in model size, we also underline the best value among models with fewer than 2M activated parameters. Error bars indicate the difference between the maximum and minimum values across 10 random seeds used for CV.

| | > 100M ACTIVATED PARAM. MODELS | | | < 2M ACTIVATED PARAM. MODELS | | | | |
|---|---|---|---|---|---|---|---|---|
| | ENFORMER (252M) | DNABERT-2 (117M) | NT-v2 (500M) | HYENADNA (1.6M) | CADUCEUS-PH (1.9M) | CONVNOVA (1.7M) | JANUSDNA MLP W/ MIDATTN (2.001M) | JANUSDNA MLP W/O MIDATTN (2.009M) |
| *Histone Markers* | | | | | | | | |
| H3 | 0.719±0.048 | 0.785±0.033 | 0.784±0.047 | 0.779±0.037 | 0.815±0.048 | 0.812±0.017 | **0.835**±0.009 | 0.831±0.023 |
| H3K14AC | 0.288±0.077 | 0.516±0.028 | 0.551±0.021 | 0.612±0.065 | 0.631±0.026 | 0.644±0.009 | **0.729**±0.022 | 0.718±0.026 |
| H3K36ME3 | 0.344±0.055 | 0.591±0.020 | 0.625±0.013 | 0.613±0.041 | 0.601±0.129 | 0.661±0.019 | 0.702±0.015 | 0.699±0.025 |
| H3K4ME1 | 0.291±0.061 | 0.511±0.028 | 0.550±0.021 | 0.512±0.024 | 0.523±0.039 | 0.554±0.023 | 0.615±0.035 | **0.616**±0.018 |
| H3K4ME2 | 0.211±0.069 | 0.336±0.040 | 0.319±0.045 | 0.455±0.095 | 0.487±0.170 | 0.485±0.032 | **0.589**±0.023 | 0.586±0.019 |
| H3K4ME3 | 0.158±0.072 | 0.352±0.077 | 0.410±0.033 | 0.549±0.056 | 0.544±0.045 | 0.566±0.027 | **0.688**±0.026 | 0.675±0.014 |
| H3K79ME3 | 0.496±0.042 | 0.613±0.030 | 0.626±0.026 | 0.672±0.048 | 0.697±0.077 | 0.700±0.007 | **0.747**±0.013 | 0.743±0.009 |
| H3K9AC | 0.420±0.063 | 0.542±0.029 | 0.562±0.040 | 0.581±0.061 | 0.622±0.030 | 0.658±0.011 | **0.673**±0.014 | 0.661±0.027 |
| H4 | 0.732±0.076 | 0.796±0.027 | 0.799±0.025 | 0.763±0.044 | 0.811±0.022 | 0.808±0.008 | 0.812±0.011 | **0.813**±0.013 |
| H4AC | 0.273±0.063 | 0.463±0.041 | 0.495±0.032 | 0.564±0.038 | 0.621±0.054 | 0.636±0.011 | 0.698±0.013 | **0.705**±0.023 |
| *Regulatory Annotation* | | | | | | | | |
| ENHANCER | 0.451±0.108 | 0.516±0.098 | 0.548±0.144 | 0.517±0.117 | 0.546±0.073 | **0.586**±0.038 | 0.559±0.042 | 0.542±0.044 |
| ENHANCER TYPES | 0.309±0.134 | 0.423±0.051 | 0.424±0.132 | 0.386±0.185 | 0.439±0.054 | 0.500±0.018 | **0.503**±0.038 | 0.492±0.096 |
| PROMOTER: ALL | 0.954±0.006 | 0.971±0.006 | **0.976**±0.006 | 0.960±0.005 | 0.970±0.004 | 0.967±0.001 | 0.970±0.002 | 0.970±0.003 |
| NONTATA | 0.955±0.010 | 0.972±0.005 | **0.976**±0.005 | 0.959±0.008 | 0.969±0.011 | 0.968±0.003 | 0.971±0.004 | 0.971±0.003 |
| TATA | 0.960±0.023 | 0.955±0.021 | 0.966±0.013 | 0.944±0.040 | 0.953±0.016 | **0.969**±0.003 | 0.958±0.007 | 0.960±0.008 |
| *Splice Site Annotation* | | | | | | | | |
| ALL | 0.848±0.019 | 0.939±0.009 | **0.983**±0.008 | 0.956±0.011 | 0.940±0.027 | 0.965±0.004 | 0.967±0.005 | 0.943±0.020 |
| ACCEPTOR | 0.914±0.028 | 0.975±0.006 | **0.981**±0.011 | 0.958±0.010 | 0.937±0.033 | 0.971±0.003 | 0.957±0.012 | 0.961±0.009 |
| DONOR | 0.906±0.027 | 0.963±0.006 | **0.985**±0.022 | 0.949±0.024 | 0.948±0.025 | 0.965±0.003 | 0.948±0.008 | 0.935±0.016 |

#### 4.2.2 Nucleotide Transformer Tasks

Next, we evaluate our model on the Nucleotide Transformer tasks, which include 18 datasets covering histone marker prediction, regulatory annotation prediction, and splice site annotation prediction. Following the evaluation metrics outlined in [11], we perform 10-fold cross-validation for each task, adhering to the same experimental settings as [8].

As shown in Table 2, our model achieves state-of-the-art performance on 12 out of 18 tasks, surpassing previous models, including those with 250 times more activated parameters. While the promoter and splice site annotation tasks exhibit slightly weaker performance compared to the best larger model, this underscores the potential importance of training data scale and diversity for these specific tasks. For clarity, we present only the results of Caduceus-PH in the table due to space constraints, as Caduceus-PS performs slightly worse than Caduceus-PH.

#### 4.2.3 DNA Long Range Benchmark

To further assess our model's ability to capture long-range dependencies in DNA sequences, we evaluate it on the expression Quantitative Trait Loci (eQTL) prediction task from DNALONGBENCH [38] with the sequence length of 450,000. The eQTL task measures whether a nucleotide variant can influence the expression of a target gene based on the sequences of the gene and its surrounding regions.

Table 3: DNALongBench eQTL Tasks. The AUROC for expert model - Enformer, Caduceus-PH, and JanusDNA. The best results are **bolded**.

| MODELS ACTIVATED PARAM | EXPERT MODEL (252M) | CADUCEUS-PH (7.7M) | JANUSDNA W/O MID-ATTN (7.662M) | JANUSDNA MLP W/O MID-ATTN (7.745M) |
|---|---|---|---|---|
| ARTERY TIBIAL | 0.741 | 0.690 | 0.803 | **0.852** |
| ADIPOSE SUBCUTANEOUS | 0.736 | 0.759 | 0.741 | **0.769** |
| CELLS CULTURED FIBROBLASTS | 0.639 | 0.690 | 0.771 | **0.802** |
| MUSCLE SKELETAL | 0.621 | 0.789 | 0.803 | **0.864** |
| NERVE TIBIAL | 0.683 | 0.842 | 0.877 | **0.914** |
| SKIN NOT SUN EXPOSED SUPRAPUBIC | 0.710 | 0.812 | 0.875 | **0.903** |
| SKIN SUN EXPOSED LOWER LEG | 0.700 | 0.692 | 0.706 | **0.846** |
| THYROID | 0.612 | 0.703 | 0.752 | **0.793** |
| WHOLE BLOOD | 0.689 | 0.769 | 0.794 | **0.821** |

Due to limited computational resources, we compare our model against the state-of-the-art DNA language model, Caduceus-PH [8], which is also trained with the same data scale for fair comparison, and the expert model for this task, Enformer [39]. The results for Enformer are taken directly from DNALONGBENCH [38]. For Caduceus-PH, we entirely fine-tune the official HuggingFace-released weights, which are pretrained on sequences of length 131k. Our model is pretrained and entirely fine-tuned using the same setup and sequence length as Caduceus-PH to ensure a fair comparison. As shown in Table 3, JanusDNA achieves the best performance on 8 out of 9 datasets, significantly outperforming the expert model and Caduceus-PH despite using fewer computational parameters.

# 5    Conclusion

**Summary**    In this work, we introduced a novel global modeling paradigm for bidirectional DNA sequence representation, combining the bidirectional capability of masked language modeling with the speed and optimization benefits of autoregressive approaches. We proposed JanusDNA, a Mamba MoE-based DNA foundation model with global attention that enhances genomic sequence understanding while maintaining low memory complexity, supporting the processing of up to 1 million base pairs (1 Mbp) on a single 80GB GPU. Experimental results demonstrate that JanusDNA outperforms HyenaDNA, Caduceus, and other Transformer-based models across a range of benchmark tasks and the expert model on eQTL tasks. By leveraging global attention mechanisms and efficient long-range sequence processing, JanusDNA offers a powerful framework for advancing research on long-range genomic interactions.

**Limitations and Future Work**    Although JanusDNA demonstrates high learning efficiency on a fixed data scale, current training is restricted to the human reference genome for fair architectural comparison. Expanding the corpus to include human genomic variants (e.g., 1000 Genomes Project) and non-human species (e.g., primates) could further boost modeling capacity and biological insight. JanusDNA also lacks integration of multimodal data such as epigenetic states (e.g., chromatin accessibility, histone modifications) and single-cell transcriptomic profiles, which are vital for resolving cell-type-specific regulation and predicting chromatin-influenced phenotypes. Future work will incorporate these modalities and explore functional roles of key genomic features (e.g., CTCF-mediated chromatin loops, enhancer RNAs, non-coding risk variants). Experimental validation (e.g., CRISPR, organoids) will prioritize therapeutic targets, while clinical collaborations will evaluate JanusDNA's utility in personalized diagnostics and drug discovery. These efforts aim to evolve JanusDNA into a unified framework linking genome structure, epigenetic regulation, and disease mechanisms.

# 6    Acknowledgments

The authors acknowledge the Scientific Computing of the IT Division at the Charité - Universitätsmedizin Berlin and the Science Computing at Shenzhen Technology University for providing computational resources that have contributed to the research results reported in this paper. B.W. and R.E. acknowledge support by the Collaborative Research Center (SFB 1470) funded by the German Research Council (DFG).

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

# A  Related Work

## A.1  DNA Language Models

**Attention-based Models**  Attention-based DNA language models, such as DNABERT [9], DNABERT2 [10], and Nucleotide Transformer [11], have demonstrated significant success in modeling DNA sequences. These models employ k-mer encoders to group consecutive nucleotide base pairs into single tokens, enabling efficient sequence representation. However, their global attention mechanisms restrict scalability to sequences of approximately 12,000 base pairs (bps), limiting their ability to capture long-range dependencies. Furthermore, the use of k-mer tokenizers reduces modeling resolution, posing challenges for tasks like Single Nucleotide Polymorphism (SNP) analysis.

**State Space Models**  State Space Models (SSMs) have been applied to genomic tasks, with HyenaDNA [15] utilizing the Hyena operator [41] to process sequences up to 1 million base pairs (bps). Despite this capability, its unidirectional design limits the model's ability to capture bidirectional genomic contexts [29]. To overcome this, Caduceus [8] introduces a bidirectional Mamba architecture, which aggregates information from both upstream and downstream sequences, enhancing genomic context comprehension. However, SSM-based models often face challenges in memory recall tasks when compared to transformer-based approaches [42]. Additionally, their masking-based training paradigm is less efficient, as it uses only 15% of the data for loss computation per iteration. In contrast, autoregressive training paradigms take advantage of nearly 99% of the data, significantly improving training efficiency and overall performance. Latest SSM-based DNA models, such as Evo [19] and Evo2 [20], introduce stripedHyena and StripedHyena2, showing better scaling rate and improved throughput compared to HyenaDNA. However, to gain better generative performance, they still rely on the autoregressive training paradigm, facing the same limitation as HyenaDNA for bidirectional understanding.

**Hybrid Models**  Hybrid models incorporate multiple encoding mechanisms such as convolutional neural networks (CNNs), Mamba, and Attention to leverage the complementary strengths of each architecture. Enformer [39] combines CNNs with Transformers, enabling the model to capture long-range genomic interactions spanning up to 100 kilobases. HybriDNA [43] integrates Mamba and Transformer components, extending its receptive field to 131 kilobases.

# B  Experimental Details

## B.1  Ablation Experiments

### B.1.1  JanusDNA Hybrid Architecture

To determine the optimal hybrid architecture for DNA sequence modeling that effectively balances local and global attention, we perform ablation experiments on various configurations of the unidirectional encoder (i.e., modules preceding the bidirectional fusion layer). Referring to [24], we also explore the value of the additional mid-attention layer. Specifically, we evaluate the following configurations: 1) mamba and FFN blocks only, 2) mamba and FFN blocks with mid-attention, 3) mamba and FFN blocks with MoE, and 4) mamba and FFN blocks with both mid-attention and MoE. The ratio of MoE to replace FFNs is set to 0.5. In models with mid-attention, the two independent bidirectional Mamba blocks at the 4th layer are replaced with two independent causal Attention blocks implemented using FlashAttention2 [44].

The models are pre-trained on sequences of lengths 1024 and 131072, with batch sizes of 128 and 1, respectively, using a single GPU. All other hyperparameters and training settings are consistent with the pre-training setup described earlier. To ensure a consistent number of activated parameters across different models, we adjust the model configurations as Table 4. The training perplexity results for these configurations are shown in Figure 5.

The training perplexity results reveal that the model with mamba, FFN, and mid-attention exhibits higher perplexity compared to the model with only mamba and FFN, while the model with mamba, FFN, and MoE achieves lower perplexity. This suggests that mid-attention may not enhance training efficiency, whereas MoE contributes positively. Notably, the model combining mamba, FFN, mid-

Table 4: Hyperparameter settings for JanusDNA ablation experiments.

| | JANUSDNA (ALL WITH MAMBA) | | | |
|---|---|---|---|---|
| | FFN | MIDATTN+FFN | MOE+FFN | MIDATTN+MOE+FFN |
| LAYERS | 8 | 8 | 8 | 8 |
| WIDTH | 148 | 148 | 128 | 128 |
| ACTIVATED PARAMS (M) | 5.973 | 5.973 | 6.084 | 6.080 |
| TOTAL PARAMS (M) | 5.973 | 5.973 | 28.104 | 28.100 |
| GLOBAL STEPS | 10K | 10K | 10K | 10K |
| EXPERT NUMBER OF MoE | 0 | 0 | 16 | 16 |
| HEAD NUMBER OF ATTENTION | 4 | 4 | 4 | 4 |
| MULTIPLE NUMBER OF FFN WIDTH | 4 | 4 | 4 | 4 |
| OPTIMIZER | ADAMW | | | |
| OPTIMIZER MOMENTUM | $\beta_1, \beta_2 = 0.9, 0.95$ | | | |
| LEARNING RATE | $8e^{-3}$ | | | |
| LR SCHEDULER | COSINE DECAY | | | |
| WEIGHT DECAY (MODEL) | 0.1 | | | |

Figure 5: Training perplexity of Mid-attention and MoE ablation models on 1024-length and 131k-length sequences.

attention, and MoE achieves the lowest perplexity, indicating that mid-attention and MoE can synergistically improve training performance.

To further investigate the role of mid-attention, we conduct additional ablation experiments comparing models with and without mid-attention (all incorporating MoE due to its demonstrated benefits) on the Nucleotide Transformer Benchmark with 3-fold cross-validation since Genomic Benchmark is too saturated to show apparent differences. We pretrain and fine-tune models with hidden dimensions of 32, 72, and 128. The results, summarized in Figure

6, indicate that models with mid-attention perform better at a hidden dimension of 32. However, as the hidden dimension increases, models without mid-attention outperform those with mid-attention at a dimension of 72. At a dimension of 128, the performance of models with and without mid-attention becomes comparable. These findings suggest that mid-attention provides diminishing benefits as the hidden dimension grows larger, eventually becoming negligible at higher dimensions.

Given that larger hidden dimensions are generally preferred for large-scale DNA sequence modeling, and considering that mid-attention introduces additional computational overhead, we prefer to use models with mamba and FFN blocks without mid-attention for downstream tasks. Nonetheless, we include the experiments of the model with mid-attention in our formal evaluations on Genomic Benchmark and Nucleotide Transformer Benchmark to ensure completeness and provide a comprehensive analysis.

### B.1.2 Reverse Complement (RC)

DNA follows the complementary base-pairing principle, meaning each DNA strand has a reverse complement strand with equivalent genetic information. However, despite this theoretical equivalence, many biologically important motifs (e.g., transcription factor binding sites) are non-palindromic. Recognizing both the forward and RC versions of such motifs (for instance, motifs like GATA and TATC) is challenging, as it effectively requires the model to learn two distinct representations. Therefore, explicitly integrating RC information allows the model to more robustly and comprehensively capture DNA sequence patterns. Nonetheless, the utility of RC depends heavily on the specific biological

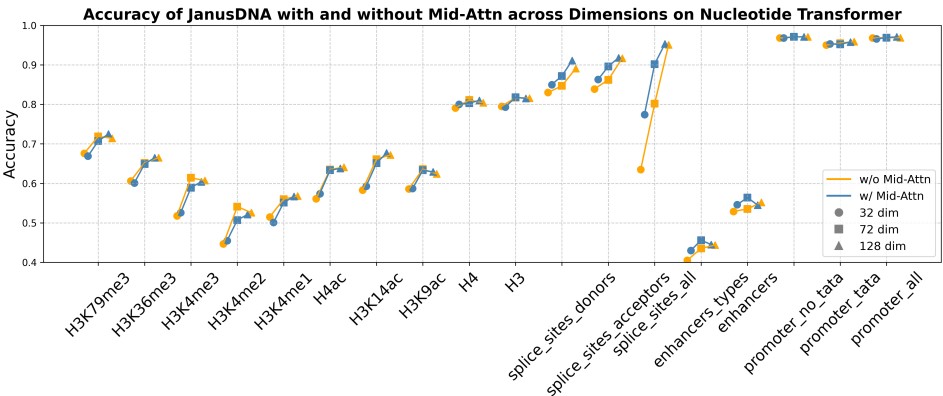

Figure 6: Mid attention ablation results on Nucleotide Transformer.

context. For instance, genomic elements such as splice sites are strictly defined by a single DNA strand, making RC inclusion potentially suboptimal or misleading.

Regarding computational overhead, RC is introduced only during inference by averaging the embedding representations of a strand and its RC prior to decoding. Thus, the additional computational cost is minimal, primarily involving an extra forward pass through the model for the RC strand.

To quantitatively assess the impact of incorporating RC, we conducted ablation experiments using the NT benchmark. We fine-tuned JanusDNA models under consistent experimental conditions (learning rate 1e-3, batch size 256), with and without RC during prediction. The results clearly indicate that models utilizing RC generally outperform those without RC across most tasks. Notably, the exception to this pattern was observed in splice site prediction tasks, where RC inclusion led to inferior performance, consistent with the biological reality that splice sites are inherently strand-specific.

Table 5: Performance of JanusDNA with and without middle attention, with and without Reverse Complement on Nucleotide Transformer Benchmark. Top-1 accuracy (↑) across 5-fold cross-validation (CV) for different model variants. Best values per task within each group (left two columns are JanusDNA without middle attention, right two columns are JanusDNA with middle attention) are **bolded**. Error bars indicate the standard deviation across 5 random seeds used for CV.

| | JANUSDNA | | | |
| | W/ MID-ATTN | | W/O MID-ATTN | |
| TASKS | W/O RC | W/ RC | W/O RC | W/ RC |
|---|---|---|---|---|
| H3 | 0.789±0.028 | **0.828**±0.020 | 0.795±0.026 | **0.830**±0.015 |
| H3K14AC | 0.689±0.029 | **0.729**±0.022 | 0.662±0.015 | **0.700**±0.015 |
| H3K36ME3 | 0.661±0.021 | **0.701**±0.022 | 0.658±0.016 | **0.688**±0.012 |
| H3K4ME1 | 0.574±0.025 | **0.609**±0.022 | 0.555±0.030 | **0.605**±0.028 |
| H3K4ME2 | 0.546±0.026 | **0.588**±0.020 | 0.532±0.020 | **0.581**±0.024 |
| H3K4ME3 | 0.640±0.013 | **0.681**±0.016 | 0.625±0.015 | **0.675**±0.014 |
| H3K79ME3 | 0.723±0.025 | **0.747**±0.013 | 0.710±0.020 | **0.743**±0.009 |
| H3K9AC | 0.638±0.023 | **0.673**±0.014 | 0.631±0.016 | **0.658**±0.020 |
| H4 | 0.781±0.020 | **0.810**±0.022 | 0.775±0.019 | **0.813**±0.011 |
| H4AC | 0.653±0.023 | **0.696**±0.019 | 0.629±0.017 | **0.684**±0.020 |
| ENHANCERS | 0.382±0.035 | **0.396**±0.033 | 0.379±0.041 | **0.397**±0.065 |
| ENHANCERSTYPES | 0.475±0.053 | **0.488**±0.066 | 0.490±0.046 | **0.492**±0.096 |
| PROMOTERALL | 0.964±0.003 | **0.969**±0.002 | 0.962±0.003 | **0.970**±0.002 |
| PROMOTERNOTATA | 0.961±0.004 | **0.969**±0.004 | 0.961±0.004 | **0.970**±0.005 |
| PROMOTERTATA | 0.946±0.012 | **0.954**±0.010 | 0.947±0.010 | **0.953**±0.019 |
| SPLICESITESALL | **0.961**±0.003 | 0.948±0.008 | **0.946**±0.007 | 0.922±0.019 |
| SPLICESITESACCEPTORS | **0.928**±0.010 | 0.906±0.016 | **0.932**±0.014 | 0.904±0.008 |
| SPLICESITESDONORS | **0.915**±0.008 | 0.893±0.006 | **0.921**±0.009 | 0.874±0.011 |

### B.1.3  Janus and Casual Modeling

We conducted an ablation comparing Masked and Janus modeling using the same architecture as in Figure 4. In order to further demonstrate the value of bidirectional modeling, we here conduct the ablation experiments between Janus modeling and Casual modeling.

Table 6: Comparasion of Janus models and Casual models on Last-Token prediction.

| | Janus | | | Casual | | |
|---|---|---|---|---|---|---|
| size(M) | 0.056 | 0.207 | 0.795 | 0.061 | 0.209 | 0.814 |
| step(k)\dim | 32 | 64 | 128 | 40 | 76 | 152 |
| 1 | 0.428 | 0.439 | 0.445 | 0.421 | 0.426 | 0.434 |
| 2 | 0.445 | 0.451 | 0.460 | 0.454 | 0.441 | 0.453 |
| 3 | 0.450 | 0.462 | 0.472 | 0.443 | 0.447 | 0.471 |
| 4 | 0.451 | 0.466 | 0.473 | 0.443 | 0.451 | 0.468 |
| 5 | 0.453 | 0.470 | 0.479 | 0.454 | 0.457 | 0.478 |
| 6 | 0.455 | 0.473 | 0.482 | 0.451 | 0.453 | 0.481 |
| 7 | 0.458 | 0.474 | 0.484 | 0.449 | 0.457 | 0.482 |
| 8 | 0.459 | 0.477 | 0.488 | 0.459 | 0.465 | 0.479 |
| 9 | 0.461 | 0.479 | 0.490 | 0.453 | 0.467 | 0.479 |
| 10 | 0.461 | 0.480 | 0.491 | 0.459 | 0.465 | 0.484 |

Table 7: Comparasion of Janus models and Casual models on Middle-Token prediction.

| | Janus | | | Casual | | |
|---|---|---|---|---|---|---|
| size(M) | 0.056 | 0.207 | 0.795 | 0.061 | 0.209 | 0.814 |
| step(k)\dim | 32 | 64 | 128 | 40 | 76 | 152 |
| 1 | 0.428 | 0.436 | 0.440 | 0.384 | 0.400 | 0.408 |
| 2 | 0.438 | 0.454 | 0.465 | 0.429 | 0.425 | 0.443 |
| 3 | 0.437 | 0.449 | 0.467 | 0.436 | 0.446 | 0.454 |
| 4 | 0.443 | 0.465 | 0.471 | 0.434 | 0.451 | 0.454 |
| 5 | 0.444 | 0.477 | 0.477 | 0.434 | 0.449 | 0.451 |
| 6 | 0.456 | 0.474 | 0.482 | 0.428 | 0.442 | 0.446 |
| 7 | 0.458 | 0.471 | 0.477 | 0.438 | 0.457 | 0.461 |
| 8 | 0.453 | 0.478 | 0.478 | 0.434 | 0.455 | 0.461 |
| 9 | 0.458 | 0.481 | 0.485 | 0.447 | 0.458 | 0.465 |
| 10 | 0.456 | 0.482 | 0.488 | 0.443 | 0.456 | 0.465 |

Conducting a direct ablation with same architecture between Casual modeling and Janus modeling is challenging due to inherent architectural differences. The core strength of Janus modeling lies in its bidirectional context understanding, requiring a bidirectional architecture. Casual modeling, by definition, is strictly unidirectional, and adapting it to a bidirectional architecture would conflict with its fundamental principles.

However, we can still maintain the most fairness by keeping models with same parameters. We constructed Casual models with one layer of a single mamba encoder and a FlashAttention2 layer with a causal attention mask, ensuring comparable model sizes to the Janus models. Following same pre-training settings, we compared the two kinds of models in last-token prediction and middle-token prediction. we expected similar performance for the prediction of the last token (where there is no bidirectional information, so both models have exactly the same information to predict the last token), whereas we expected Janus models to outperform Casual models for the prediction of a token in the center of the sequence (where Janus benefits from the bidirectional context).

The results are presented in table 6 and table 7. We confirm that the Janus model consistently outperforms the Casual models for tokens in the center of the sequence across all evaluated model sizes. To our surprise, Janus models also slightly outperform the Casual models even on the last token prediction task, suggesting Janus models indeed learn richer DNA representations through bidirectional training.

### B.1.4 Multilayer Perceptron as feature enhancer

The features fused by attention can be further enhanced through the addition of a Multi-Layer Perceptron (MLP). To verify this, we conduct ablation experiments and name the JanusDNA models equipped with an MLP as JanusDNA-MLP, distinguishing them from the original JanusDNA models without the MLP attached after the feature fusion attention module. We perform the ablation

experiments on both the Nucleotide Transformer Benchmark and the DNALONGBENCH Benchmark, as shown in Table 8 and Table 3. The inclusion of the MLP leads to notable performance improvements across both benchmarks.

Table 8: Performance of JanusDNA with and without MLP on Nucleotide Transformer Tasks. Performance (↑) across 10-fold CV for janusDNA and JanusDNA mlp variants. Metrics vary by task: MCC for histone markers and enhancer annotation, F1-score for promoter annotation and splice site acceptor/donor, and accuracy for splice site "all". Best values per task are **bolded**, second best are *italicized*. Since all models are approximately fewer than 2M activated parameters, we underline the best value(s) among them. Error bars indicate the difference between the maximum and minimum values across 10 random seeds used for CV.

| | JANUSDNA W/ MID-ATTN (1.980M) | JANUSDNA W/O MIDATTN (1.988M) | JANUSDNA MLP W/ MIDATTN (2.001M) | JANUSDNA MLP W/O MIDATTN (2.009M) |
|---|---|---|---|---|
| *Histone Markers* | | | | |
| H3 | 0.821±0.021 | 0.824±0.012 | **0.835**±0.009 | 0.831±0.023 |
| H3K14AC | 0.665±0.034 | 0.685±0.016 | **0.729**±0.022 | 0.718±0.026 |
| H3K36ME3 | 0.658±0.024 | 0.670±0.012 | **0.702**±0.015 | 0.699±0.025 |
| H3K4ME1 | 0.563±0.041 | 0.571±0.018 | 0.615±0.035 | **0.616**±0.018 |
| H3K4ME2 | 0.509±0.056 | 0.548±0.022 | **0.589**±0.023 | 0.586±0.019 |
| H3K4ME3 | 0.605±0.030 | 0.629±0.022 | **0.688**±0.026 | 0.675±0.014 |
| H3K79ME3 | 0.716±0.017 | 0.727±0.023 | **0.747**±0.013 | 0.743±0.009 |
| H3K9AC | 0.641±0.024 | 0.639±0.019 | **0.673**±0.014 | 0.661±0.027 |
| H4 | 0.809±0.021 | **0.816**±0.008 | 0.812±0.011 | 0.813±0.013 |
| H4AC | 0.637±0.060 | 0.653±0.034 | 0.698±0.013 | **0.705**±0.023 |
| *Regulatory Annotation* | | | | |
| ENHANCER | **0.564**±0.022 | 0.535±0.036 | 0.559±0.042 | 0.542±0.044 |
| ENHANCER TYPES | 0.462±0.049 | 0.470±0.025 | **0.503**±0.038 | 0.492±0.096 |
| PROMOTER: ALL | 0.969±0.002 | **0.971**±0.002 | 0.970±0.002 | 0.970±0.003 |
|   NONTATA | 0.971±0.003 | 0.971±0.002 | 0.971±0.004 | 0.971±0.003 |
|   TATA | 0.956±0.010 | 0.958±0.008 | 0.958±0.007 | **0.960**±0.008 |
| *Splice Site Annotation* | | | | |
| ALL | 0.963±0.022 | 0.960±0.009 | **0.967**±0.005 | 0.943±0.020 |
| ACCEPTOR | 0.949±0.020 | 0.939±0.022 | 0.957±0.012 | **0.961**±0.009 |
| DONOR | 0.947±0.015 | 0.936±0.014 | **0.948**±0.008 | 0.935±0.016 |

## B.2 Formal Experiment

### B.2.1 Pre-training

**Architecture configuration** We utilize Mamba, FFN, MoE blocks as the primary building blocks for unidirectional representation, which are then followed by a bidirectional fusion layer. There are 8 layers in each unidirectional encoder and each layer consists of one Mamba block and one FFN block. The MoE block is to replace the FFN block at a certain ratio, which is set to 0.5 in our experiments. The number of experts is set to 16 and the dimension of FFN is set to 4 times the hidden dimension. The bidirectional fusion layer is achieved by a FlexAttention layer [26] with 4 attention heads.

Meanwhile, we also implement a version of the model with mid-attention, which replaces the Mamba block at the fifth layer with a mid-attention layer. The mid-attention layer is implemented with FlexAttention with 4 attention heads. The attention mask is set to half triangle to allow the model to only attend to the tokens ahead of the current token to keep causality.

**Pre-training setup** To ensure a fair comparison with prior work, we pre-train our model on the human reference genome (HG38 [32]) following the training setup described in [8]. We use cross-entropy loss for pre-training. The model is trained with a learning rate of $8 \times 10^{-3}$, maintaining a constant token count of $2^{20}$ tokens per batch. Two sequence lengths are used: 1024 and 131072, with corresponding batch sizes of 128 and 1, respectively, across 8 GPUs. Optimization is performed using AdamW [45] with a weight decay of 0.1, $\beta_1 = 0.9$, and $\beta_2 = 0.95$. A cosine learning rate scheduler is applied, incorporating a warmup phase for 10% of the training steps. The learning rate starts at $1 \times 10^{-6}$ and peaks at $1 \times 10^{-4}$. The coefficient for the MoE auxiliary loss is set to 0.2. The gradient clipping threshold is set to 1.0.

For the 1024-length model, training is conducted for 10,000 steps, while the 131072-length model is trained for 50,000 steps.

We pre-trained three scales of the model: 32, 72, and 144 hidden dimensions to keep the same activated parameters for fair comparison with baseline models on different benchmarks. The hidden dimensions of 32 and 72 are used for the 1024-length model, while the 144 hidden dimension is used for the 131072-length model. The feedforward dimension is set to 4 times the hidden dimension, and the number of attention heads is set to 8. The coefficient for the MoE auxiliary loss is set to 0.2, the number of experts is set to 16. All hyperparameter settings are listed as Table 9.

Table 9: Hyperparameter settings for JanusDNA pretraining (select models).

|  | JANUSDNA | | | | |
|  | W/ MIDATTN | | W/O MIDATTN | | |
| LAYERS | 8 | 8 | 8 | 8 | 8 |
| WIDTH | 32 | 72 | 32 | 72 | 144 |
| ACTIVATED PARAMS (M) | 0.422 | 1.980 | 0.428 | 1.989 | 7.664 |
| TOTAL PARAMS (M) | 1.798 | 8.947 | 1.804 | 8.956 | 35.533 |
| MAX SEQ. LEN. | 1024 | 1024 | 1024 | 1024 | 131072 |
| BATCH SIZE | 1024 | 1024 | 1024 | 1024 | 8 |
| GLOBAL STEPS | 10K | 10K | 10K | 10K | 50K |
| EXPERT NUMBER OF MoE | 16 | 16 | 16 | 16 | 16 |
| HEAD NUMBER OF ATTENTION | 4 | 4 | 4 | 4 | 4 |
| MULTIPLE NUMBER OF FFN WIDTH | 4 | 4 | 4 | 4 | 4 |
| COEFFICIENT OF AUXILIARY MoE LOSS | 0.2 | 0.2 | 0.2 | 0.2 | 0.2 |
| RUNTIME(H800 WITH EVALUATION EVERY 2K STEPS) | 3H3M | 3H7M | 3H2M | 3H8M | 9H17M |
| OPTIMIZER | ADAMW | | | | |
| OPTIMIZER MOMENTUM | $\beta_1, \beta_2 = 0.9, 0.95$ | | | | |
| LEARNING RATE | $8e^{-3}$ | | | | |
| LR SCHEDULER | COSINE DECAY | | | | |
| WEIGHT DECAY (MODEL) | 0.1 | | | | |

## B.2.2 Benchmarks

**Genomic Benchmark** For the Genomic Benchmark tasks, we follow the experimental setup of [8] and report their results for comparison. To ensure a fair evaluation, we fine-tune 32-dimensional models to match the activated parameter count of the baselines.

We apply 5-fold cross-validation, splitting the training set into 90/10 train/validation splits and using early stopping based on validation performance with seeds of $\{1, 2, 3, 4, 5\}$. Models are fine-tuned for 10 epochs with a batch size of 256.

For learning rate selection, we perform hyperparameter tuning over $1 \times 10^{-3}, 2 \times 10^{-3}$ as [8], and report the best-performing configuration across cross-validation, as summarized in Table 10. We use cross-entropy loss for fine-tuning. For JanusDNA, the coefficient for the MoE auxiliary loss is set to 0.2.

Table 10: JanusDNA models with and without mid-attention hyperparameter selection for learning rate on genomic benchmarks for top-1 accuracy averaged over 5-fold cross-validation.

| DATASET | W/ MIDATTN | W/O MIDATTN |
|---|---|---|
| MOUSE ENHANCERS | $1e^{-3}$ | $2e^{-3}$ |
| CODING VS. INTERGENOMIC | $1e^{-3}$ | $1e^{-3}$ |
| HUMAN VS. WORM | $1e^{-3}$ | $1e^{-3}$ |
| HUMAN ENHANCERS COHN | $1e^{-3}$ | $1e^{-3}$ |
| HUMAN ENHANCER ENSEMBL | $1e^{-3}$ | $1e^{-3}$ |
| HUMAN REGULATORY | $1e^{-3}$ | $1e^{-3}$ |
| HUMAN OCR ENSEMBL | $1e^{-3}$ | $2e^{-3}$ |
| HUMAN NONTATA PROMOTERS | $1e^{-3}$ | $1e^{-3}$ |

**Nucleotide Transformer Tasks** For the Nucleotide Transformer tasks, we adopt the experimental setup from [8] and report their results for comparison. To ensure a fair comparison, we fine-tune 72-dimensional models to match the activated parameter count of the baseline models.

We use 10-fold cross-validation with seeds of $\{1, 2, 3, 4, 5, 6, 7, 8, 9, 10\}$, splitting the dataset into 90/10 train/validation subsets and applying early stopping based on validation performance. Each model is fine-tuned for 20 epochs.

We use cross-entropy loss for fine-tuning. We conduct hyperparameter tuning over the following search space, the same as [8]: learning rates in $1 \times 10^{-3}, 2 \times 10^{-3}$ and batch sizes in $128, 256, 512$. For each task, we report the best-performing configuration across cross-validation, as summarized in Table 11. For JanusDNA, the coefficient for the MoE auxiliary loss is set to 0.2.

Table 11: JanusDNA Hyperparameter Selection for Nucleotide Transformer Tasks. Model with Mid-Attention and Model without Mid-Attention fine-tuning hyperparameters chosen based on best performance averaged over 10-fold cross-validation.

| | | JanusDNA | | | |
| --- | --- | --- | --- | --- | --- |
| | | W/ MIDATTN | | W/O MIDATTN | |
| | | LR | BATCH SIZE | LR | BATCH SIZE |
| HISTONE MARKERS | H3 | $1e^{-3}$ | 256 | $2e^{-3}$ | 128 |
| | H3K14AC | $1e^{-3}$ | 256 | $1e^{-3}$ | 256 |
| | H3K36ME3 | $1e^{-3}$ | 256 | $2e^{-3}$ | 512 |
| | H3K4ME1 | $1e^{-3}$ | 256 | $1e^{-3}$ | 256 |
| | H3K4ME2 | $1e^{-3}$ | 256 | $1e^{-3}$ | 256 |
| | H3K4ME3 | $1e^{-3}$ | 256 | $1e^{-3}$ | 256 |
| | H3K79ME3 | $1e^{-3}$ | 256 | $1e^{-3}$ | 256 |
| | H3K9AC | $1e^{-3}$ | 256 | $1e^{-3}$ | 128 |
| | H4 | $1e^{-3}$ | 256 | $1e^{-3}$ | 256 |
| | H4AC | $1e^{-3}$ | 256 | $1e^{-3}$ | 256 |
| REGULATORY ANNOTATION | ENHANCERS | $1e^{-3}$ | 512 | $1e^{-3}$ | 512 |
| | ENHANCERS TYPES | $2e^{-3}$ | 256 | $2e^{-3}$ | 256 |
| | PROMOTER ALL | $1e^{-3}$ | 512 | $1e^{-3}$ | 128 |
| | PROMOTER NO TATA | $1e^{-3}$ | 256 | $1e^{-3}$ | 128 |
| | PROMOTER TATA | $2e^{-3}$ | 256 | $1e^{-3}$ | 128 |
| SPLICE SITE ANNOTATION | SPLICE SITES ACCEPTORS | $2e^{-3}$ | 128 | $2e^{-3}$ | 128 |
| | SPLICE SITES ALL | $2e^{-3}$ | 128 | $2e^{-3}$ | 128 |
| | SPLICE SITES DONORS | $2e^{-3}$ | 128 | $2e^{-3}$ | 128 |

**DNALONGBENCH**  For the DNALONGBENCH eQTL tasks, we compare JanusDNA with both the expert model Enformer and the state-of-the-art architecture Caduceus-PH. We obtain the pre-trained weights of Caduceus-PH from Hugging Face: `https://huggingface.co/kuleshov-group/caduceus-ph_seqlen-1k_d_model-256_n_layer-4_lr-8e-3`.

For all models, we extract the hidden state embeddings from the final layer and apply a pooling layer to obtain a fixed-length representation for each input sequence. A linear classification head is then used to map these representations to the target number of classes for each cell type.

To ensure comparability, we fine-tune the JanusDNA model with 144-dimensional embeddings, matching the activated parameter count of Caduceus-PH. Fine-tuning is conducted for 3 epochs using a learning rate of $4 \times 10^{-4}$ and a batch size of 8. We use cross-entropy loss for fine-tuning, and for JanusDNA, the coefficient for the MoE auxiliary loss is set to 0.02. Training is distributed across eight 80GB GPUs, with each GPU processing one batch. All models are fine-tuned and evaluated using float32 precision to ensure stability and fairness in comparison. Due to limited computational resources, we conduct a single run per sub-dataset using the same set of hyperparameters across all experiments.

**Transcription Factor Prediction (Mouse)**  We conducted experiments on non-human Genome Understanding Evaluation (GUE) tasks [10], focusing on transcription factor binding site prediction in mouse genomes to evaluate its generalization capability.

We followed DNABERT2's experimental setup, performing standard fine-tuning with a batch size of 32. Unlike DNABERT2's 1000-epoch fine-tuning at a learning rate of 3e-5, we fine-tuned the JanusDNA model for only 10 epochs at a learning rate of 1e-3.

Results, shown in Table 12, it is very interesting to see that JanusDNA, with fewer active parameters, can perform similarly to DNABERT2 on these tasks, despite being pre-trained only on the human reference genome. In future work, we plan to explore further how more diverse pre-training data affects the model performance.

Table 12: Transcription Factor Prediction (Mouse). Performance ($\uparrow$) across various models. Metrics are MCC for different categories (0-4). Best values per category are **bolded**. Given the disparity in model size, we also underline the best value among models with fewer than 100M activated parameters.

| | | TRANSCRIPTION FACTOR PREDICTION (MOUSE) | | | | |
|---|---|---|---|---|---|---|
| MODEL | ACTIVATED PARAMS | 0 | 1 | 2 | 3 | 4 |
| *> 100M activated Param. Models* | | | | | | |
| DNABERT-2 (PRE-TRAINED ON GUE) | 117M | **0.642** | **0.863** | 0.813 | 0.735 | 0.508 |
| DNABERT-2 (NOT PRE-TRAINED ON GUE) | 117M | 0.568 | 0.848 | 0.793 | 0.665 | **0.527** |
| NT-500M-HUMAN | 480M | 0.310 | 0.750 | 0.617 | 0.292 | 0.293 |
| NT-500M-1000G | 480M | 0.393 | 0.755 | 0.647 | 0.331 | 0.340 |
| NT-2500M-1000G | 2537M | 0.483 | 0.800 | 0.701 | 0.423 | 0.434 |
| NT-2500M-MULTI | 2537M | 0.633 | 0.838 | 0.715 | 0.694 | 0.471 |
| *< 100M activated Param. Models* | | | | | | |
| DNABERT (3-MER) | 86M | 0.423 | 0.791 | 0.699 | 0.554 | 0.420 |
| DNABERT (4-MER) | 86M | 0.494 | 0.800 | 0.726 | 0.518 | 0.441 |
| DNABERT (5-MER) | 87M | 0.425 | 0.793 | 0.622 | 0.499 | 0.403 |
| DNABERT (6-MER) | 89M | 0.444 | 0.789 | 0.714 | 0.449 | 0.425 |
| JANUSDNA-72DIM | 2.009M | 0.619 | 0.850 | **0.875** | **0.843** | 0.502 |

## B.3 Details of resources used

We use 80GB NVIDIA H100, A100, A800 GPUs for pre-training and fine-tuning.

