# OpenReview forum: "JanusDNA: A Powerful Bi-directional Hybrid DNA Foundation Model"
_NeurIPS.cc/2025/Conference — NeurIPS 2025 poster_

### Official Review · Reviewer_BnYd · 2025-07-01

**Clarity:** 3
**Significance:** 3
**Originality:** 3
**Rating:** 5
**Confidence:** 4

**Summary:**

This paper proposes JanusDNA, a hybrid SSM-transformer MoE architecture applied to genomics foundation models. JanusDNA achieves competitive or SOTA performance across a wide array of tasks such as the genomics benchmark, NT benchmark, and DNA long range benchmark. A distinction of JanusDNA from previous approaches is that it leverages the forward and reverse reads of genomics sequencing to create a new type of inductive biases, which are shown by the authors to be more efficient than uni-directional autoregression and MLM.

**Questions:**

NA

**Ethical Concerns:**

["NO or VERY MINOR ethics concerns only"]

**Paper Formatting Concerns:**

The format is appropriate

**Quality:**

4

**Strengths And Weaknesses:**

### __Strengths__

- Genomics foundation model is an important topic that has received a lot of attention. Having an additional model that contribute to this space is a plus.
- Unlike natural language, genomics sequences sourced in the wild may not attain a canonical order (i.e. they may be read from left to right or right to left). Thus, bidirectional attention certainly has a better inductive bias for genomics modeling, but as numerous research in scaling language models have shown, MLM does not scale well compared to autoregressive models. I believe the two-stream design for forward & backward reads is appropriate in this context.
- Orthogonal to the design of inductive bias, the authors have explored an array of best practices in scalable foundation model variants, most notably, the MoE architecture and hybrid SSM-attention models. While these improvements are not novel, to this reviewer's knowledge this paper is the first in investigating their application in genomics modeling.
- Janus-DNA achieves competitive results across a wide array of genomics modeling tasks.

### __Weaknesses__

- While I agree that the choice of dual-stream autoregression is appropriate, it'd be good to include a discussion about this point to contextualize this choice _for genomics modeling_.
- As the authors have discussed, Janus-DNA is only pretrained on Human Reference Genome and may lack the knowledge needed for modeling other species, but it'd be good to see some performance on these tasks (such as EPM and Mouse from GUE) to understand the performance gap.

---

> ### Author Rebuttal · Authors · 2025-07-30
>
> Dear Reviewer BnYd,
>
> We sincerely express our gratitude for your recognition of our work and your insightful, detailed feedback. We have carefully formulated the following responses:
>
> **W1: While I agree that the choice of dual-stream autoregression is appropriate, it'd be good to include a discussion about this point to contextualize this choice *for genomics modeling*.**
>
> Thank you very much for raising this important point. We are very thankful for the help in making this statement more clear.
>
> Autoregressive modeling relies on unidirectional representations, where understanding and predicting a current pattern depends on preceding ones. This approach is extensively applied in Natural Language Processing (NLP), as the logical sequence of human language allows subsequent elements to be inferred from prior context. Such modeling naturally aligns with language generation tasks. However, DNA, as a complex biological "language," exhibits distinct patterns significantly different from human languages in regulating phenotypes. Unlike human languages, genomic processes frequently involve bidirectional interactions, with critical regulatory elements located both upstream and downstream of key genomic regions.
>
> For example, bidirectional promoters initiate transcription in both orientations [1, 2]. Additionally, intergenic enhancers transcribed predominantly bidirectionally often function as weak promoters in both directions. Conversely, for elements with unidirectional transcription (both enhancers and promoters), transcription direction typically correlates with the orientation in which the element functions as a promoter in vivo [3].
>
> Thus, bidirectional modeling is essential for comprehensively understanding genomic processes.
>
> In addition to dual-stream autoregression, masked modeling is a classic method for bidirectional sequence modeling. However, due to its masking logic, only a subset of tokens contribute to model optimization, as discussed in our "Empirical Validation of Training Efficiency" section (Section 3.1).
>
> Therefore, Janus modeling, the dual-stream autoregressive method proposed in our work, facilitates effective bidirectional DNA modeling while maintaining high learning efficiency, making it particularly suitable for genomic modeling.
>
> We briefly addressed this topic in Section 1 (lines 55–60). Incorporating the above discussion will further enhance clarity and biological relevance. We will include this expanded content in the camera-ready version, as you suggested. Thank you again for highlighting this important point.
>
> [1] Wei, Wu, et al. "Functional consequences of bidirectional promoters." Trends in Genetics 27.7 (2011): 267-276.
>
> [2]Schulz, Daniel, et al. "Transcriptome surveillance by selective termination of noncoding RNA synthesis." Cell 155.5 (2013): 1075-1087.
>
> [3]Mikhaylichenko, Olga, et al. "The degree of enhancer or promoter activity is reflected by the levels and directionality of eRNA transcription." Genes & development 32.1 (2018): 42-57.
>
> **W2: As the authors have discussed, Janus-DNA is only pretrained on Human Reference Genome and may lack the knowledge needed for modeling other species, but it'd be good to see some performance on these tasks (such as EPM and Mouse from GUE) to understand the performance gap.**
>
> Thank you very much for your interest in exploring further possibilities for our work; we greatly appreciate it. Currently, we are integrating genomic data from additional species, variants, and incorporating modalities, such as epigenetic states (e.g., chromatin accessibility, histone modifications) and single-cell transcriptomic profiles. These additions are vital for understanding cell-type-specific regulation and predicting chromatin-influenced phenotypes within the JanusDNA framework.
>
> We fully agree that evaluating models pre-trained on the human reference genome across other species to determine performance gaps will help explore genomic similarities between species and contribute to future biological research. We will pursue this direction in our future work. We express our sincere gratitude for your insightful suggestions and the valuable research ideas proposed.
>
>
> Thank you once again for your valuable feedback. Your comments significantly enhance the quality of our manuscript, and we are committed to making the necessary improvements to meet the high standards of the NeurIPS community.

---

> > ### Comment · Reviewer_BnYd · 2025-08-04
> >
> > I'd like to thank the authors for their feedback. I would still like to see more experiments on non-human genomics to have a better understanding of the model's performance, though! I also encourage the authors to include the discussions on the choice of dual-stream autoregression in their paper.

---

> > > ### Author Response · Authors · 2025-08-06
> > > **Transcription Factor Binding Site Prediction in Mouse Genomes**
> > >
> > > Dear Reviewer BnYd,
> > >
> > > Thank you for your valuable suggestion to explore further performance. We conducted experiments on non-human Genome Understanding Evaluation (GUE) tasks, focusing on **transcription factor binding site prediction in mouse genomes**.
> > >
> > > We followed DNABERT2’s experimental setup, performing standard fine-tuning with a batch size of 32. Unlike DNABERT2’s 1000-epoch fine-tuning at a learning rate of 3e-5, we fine-tuned the JanusDNA model for only 10 epochs at a learning rate of 1e-3.
> > >
> > > Results, shown in the table below, it is very interesting to see that JanusDNA, with fewer active parameters, can perform similarly to DNABERT2 on these tasks, despite being pre-trained only on the human reference genome. In future work, we plan to explore further how more diverse pre-training data affects the model performance.
> > >
> > > We appreciate your constructive feedback. We will include these results and a discussion on dual-stream autoregression in the final manuscript. Thank you again!
> > >
> > > |  |  |  | Transcription Factor Prediction (Mouse)  |  |  |  |  |
> > > | --- | --- | --- | --- | --- | --- | --- | --- |
> > > | Model |  | activated params | 0 | 1 | 2 | 3 | 4 |
> > > | DNABERT (3-mer) |  | 86M | 42.31 | 79.1 | 69.9 | 55.4 | 41.97 |
> > > | DNABERT (4-mer) |  | 86M | 49.42 | 79.95 | 72.62 | 51.79 | 44.13 |
> > > | DNABERT (5-mer) |  | 87M | 42.45 | 79.32 | 62.22 | 49.92 | 40.34 |
> > > | DNABERT (6-mer) |  | 89M | 44.42 | 78.94 | 71.44 | 44.89 | 42.48 |
> > > | NT-500M-human |  | 480M | 31.04 | 75.04 | 61.67 | 29.17 | 29.27 |
> > > | NT-500M-1000g |  | 480M | 39.26 | 75.49 | 64.7 | 33.07 | 34.01 |
> > > | NT-2500M-1000g |  | 2537M | 48.31 | 80.02 | 70.14 | 42.25 | 43.4 |
> > > | NT-2500M-multi |  | 2537M | 63.31 | 83.76 | 71.52 | 69.44 | 47.07 |
> > > | DNABERT-2 | pre-trained on GUE | 117M | **64.23** | **86.28** | 81.28 | 73.49 | 50.8 |
> > > | DNABERT-2 | not pre-trained on GUE | 117M | 56.76 | 84.77 | 79.32 | 66.47 | **52.66** |
> > > | JanusDNA-72dim |  | 2.009M (8.976 M in total with 6.967 M unactivated) | 61.8954598903656 | 84.973192 | **87.466168** | **84.303385** | 50.191247 |

---

### Official Review · Reviewer_tdwT · 2025-07-03

**Clarity:** 4
**Significance:** 3
**Originality:** 3
**Rating:** 5
**Confidence:** 4

**Summary:**

This paper introduces JanusDNA, a novel bidirectional DNA foundation model that addresses key limitations in current genomic language models. The authors propose a new pretraining paradigm called "Janus modeling" that combines the training efficiency of autoregressive models with the bidirectional understanding capabilities of masked language models. The model architecture employs a hybrid Mamba-Attention-Mixture-of-Experts (MoE) design, enabling it to process sequences up to 1 million base pairs at single-nucleotide resolution on a single GPU. The model is evaluated on three genomic benchmarks, achieving state-of-the-art performance on many tasks while using significantly fewer parameters than competing models.

**Questions:**

1- Can the authors provide a more detailed explanation of why Janus modeling is nearly twice as fast as masked modeling? This seems counterintuitive given the bidirectional processing. Did you ensure both models used the same architecture for fair comparison?

2- Could you provide an ablation study comparing MLM, CLM, and Janus modeling objectives using the same model architecture? This would help isolate the contribution of your proposed training paradigm.

3- The paper mentions processing both forward and reverse complement strands. What is the computational overhead of this approach, and have you ablated its contribution to performance?

**Ethical Concerns:**

["NO or VERY MINOR ethics concerns only"]

**Final Justification:**

I have raised my score to recommend Accept. This change is a direct result of the authors' outstanding rebuttal, which convincingly addressed my primary reservations. My concerns about the model's training efficiency were clarified with a detailed architectural explanation. More importantly, the authors conducted two new, substantial ablation studies during the rebuttal period. These experiments provide the strong empirical evidence that was previously missing, first by directly comparing the Janus pre-training paradigm against a causal language model, and second by quantitatively assessing the impact of the reverse complement strategy. These additions have transformed the paper from a promising concept into a well-supported contribution.

The single remaining limitation, the pre-training on only the human reference genome, is acceptable in this context. The authors rightly justified this as a necessary choice for fair comparison with baselines in a paper focused on methodological novelty, and they have acknowledged it as an area for future work. The weight of the successfully resolved issues, backed by new data, far outweighs this reasonable scope limitation.

**Limitations:**

Yes

**Quality:**

3

**Strengths And Weaknesses:**

Strengths:
1- The Janus modeling approach is innovative, allowing each token to be predicted using its full bidirectional context while maintaining high training efficiency. This addresses a fundamental trade-off in language model pretraining.

2- The hybrid architecture combining Mamba, Attention, and MoE is novel.

3- JanusDNA achieves state-of-the-art performance on multiple benchmarks.

Weaknesses:
1- There is an inconsistency in the training efficiency claims (lines 198-199). The paper states that Janus models train "nearly twice as fast as masked models," but this seems counterintuitive given the bidirectional processing and attention mechanisms involved. The explanation for this efficiency gain is insufficient.

2- The model is only pretrained on the human reference genome (HG38) for fair comparison with baselines. This limits the model's potential and generalizability compared to models trained on more diverse genomic data.

3-  The paper would benefit from a direct comparison of different pretraining objectives (MLM, CLM, and Janus modeling) using the same architecture to isolate the contribution of the training paradigm.

---

> ### Author Rebuttal · Authors · 2025-07-30
>
> Dear reviewer tdwT,
>
> We greatly value your time and effort in providing this constructive review and recognition of our work. We carefully considered your suggestions and propose the following response for your further review:
>
> **W1&Q1: Training speed (Janus model vs masked model)**
>
> Thanks very much for pointing this out. For the Janus model, the architecture is one layer of two separated parallel unidirectional encoders followed by a FlexAttention with a specifically designed attention mask for Janus-type feature fusion. As shown in Figure 1(E) (which needs to be modified a bit that the diagonal of the left top part should be unmasked), two unidirectional features are concatenated shaped (batch, sequence_length * 2, dim) , and the attention mask of Janus fusion layer is sparse due to its design.
>
> Compared to the Janus model, the architecture of the masked model is similar, where there is one layer of two separated parallel unidirectional encoders (same as Janus model) followed by a FlexAttention, but with no attention mask. This is because for masked training, partial tokens have been masked in the dataset loading process, causing the full attention requirement for bidirectional feature fusion. Also, to keep the comparison fair, the two unidirectional features from the layer of unidirectional encoders will also be concatenated shaped (batch, sequence_length * 2, dim).
>
> Thus, the speed difference primarily arises from the attention stage, where the Janus model employs sparse attention, while the masked model uses dense, full attention. Additionally, the masking pre-processing for the masked model incurs some computational overhead. Consequently, the Janus model achieves faster performance than the masked model when the architectures are identical.
>
> Importantly, an alternative implementation of the masked model would concatenate features along the feature axis rather than the sequence axis, resulting in final attention over a tensor with shape (batch, sequence_length, dim * 2). This approach would make the computational costs of Janus and masked pre-training nearly equivalent. However, we opted against this implementation in our ablation study as concatenating along the feature axis would alter the masked model's capacity, compromising the comparability between the two approaches. We have now clarified this point in the description of the ablation experiment.
>
> **W2: limited pre-training data**
>
> We fully agree that the data scope of this work is limited to the human reference genome to achieve the fairest possible comparison with existing methods since the primary contribution of this work is the model architecture and pre-training paradigm itself as you acknowledged. We truly appreciate that. We fully agree that, for many DNA language model applications, diverse pre-training data is crucial and we are actively working on extending the data scope to genomes of additional species, more individual human genomes, and incorporating modalities such as epigenetic information and different sequencing data types to make the JanusDNA architecture more generalizable and robust. Thanks very much for your suggestion.
>
> **W3&Q2: MLM, CLM, Janus modeling ablation upon the same model architecture?**
>
> Thank you for this insightful suggestion. We conducted an ablation comparing MLM and Janus modeling using the same architecture as in Figure 3. However, conducting a direct ablation between CLM and Janus/MLM is challenging due to inherent architectural differences. The core strength of Janus modeling lies in its bidirectional context understanding, requiring a bidirectional architecture. CLM, by definition, is strictly unidirectional, and adapting it to a bidirectional architecture would conflict with its fundamental principles.
>
> Nevertheless, we acknowledge the value in exploring this further. We ran a new ablation experiment to compare the performance of a unidirectional CLM and a bidirectional Janus model. In this comparison, we expected similar performance for the prediction of the last token (where there is no bidirectional information, so both models have exactly the same information to predict the last token), whereas we expected Janus to outperform CLM for the prediction of a token in the center of the sequence (where Janus benefits from the bidirectional context). To maintain fairness, causal models were constructed with one layer of a single mamba encoder and a FlashAttention2 layer with a causal attention mask, ensuring comparable model sizes to the Janus models.
>
> The results of this new experiment are presented in the table below and will be included in the camera ready version of the manuscript. We confirm that the Janus model consistently outperforms the CLM for tokens in the center of the sequence across all evaluated model sizes. To our surprise, Janus also slightly outperforms the CLM even on the last token prediction task, suggesting Janus models indeed learn richer DNA representations through bidirectional training. We will clarify this point further in the manuscript.
>
> |  | last token |  |  |  |  |  | middle token |  |  |  |  |  |
> | --- | --- | --- | --- | --- | --- | --- | --- | --- | --- | --- | --- | --- |
> |  | janus |  |  | clm |  |  | janus |  |  | clm |  |  |
> | size(M) | 0.056 | 0.207 | 0.795 | 0.061 | 0.209 | 0.814 | 0.056 | 0.207 | 0.795 | 0.061 | 0.209 | 0.814 |
> | step(k)\dim | 32 | 64 | 128 | 40 | 76 | 152 | 32 | 64 | 128 | 40 | 76 | 152 |
> | 1 | 0.428 | 0.439 | 0.445 | 0.421 | 0.426 | 0.434 | 0.428 | 0.436 | 0.440 | 0.384 | 0.400 | 0.408 |
> | 2 | 0.445 | 0.451 | 0.460 | 0.454 | 0.441 | 0.453 | 0.438 | 0.454 | 0.465 | 0.429 | 0.425 | 0.443 |
> | 3 | 0.450 | 0.462 | 0.472 | 0.443 | 0.447 | 0.471 | 0.437 | 0.449 | 0.467 | 0.436 | 0.446 | 0.454 |
> | 4 | 0.451 | 0.466 | 0.473 | 0.443 | 0.451 | 0.468 | 0.443 | 0.465 | 0.471 | 0.434 | 0.451 | 0.454 |
> | 5 | 0.453 | 0.470 | 0.479 | 0.454 | 0.457 | 0.478 | 0.444 | 0.477 | 0.477 | 0.434 | 0.449 | 0.451 |
> | 6 | 0.455 | 0.473 | 0.482 | 0.451 | 0.453 | 0.481 | 0.456 | 0.474 | 0.482 | 0.428 | 0.442 | 0.446 |
> | 7 | 0.458 | 0.474 | 0.484 | 0.449 | 0.457 | 0.482 | 0.458 | 0.471 | 0.477 | 0.438 | 0.457 | 0.461 |
> | 8 | 0.459 | 0.477 | 0.488 | 0.459 | 0.465 | 0.479 | 0.453 | 0.478 | 0.478 | 0.434 | 0.455 | 0.461 |
> | 9 | 0.461 | 0.479 | 0.490 | 0.453 | 0.467 | 0.479 | 0.458 | 0.481 | 0.485 | 0.447 | 0.458 | 0.465 |
> | 10 | 0.461 | 0.480 | 0.491 | 0.459 | 0.465 | 0.484 | 0.456 | 0.482 | 0.488 | 0.443 | 0.456 | 0.465 |
>
> **Q3: Computational overhead of RC and ablation on its value?**
>
> Thank you very much for raising this key point. We fully agree that clarifying the computational overhead and performance impact of RC is essential.
>
> Including RC enables the model to more robustly and comprehensively capture DNA sequence patterns, especially for biologically important motifs (e.g., transcription factor binding sites) that are non-palindromic. Recognizing both forward and RC versions of such motifs (for instance, motifs like GATA and TATC) is challenging, as it effectively requires the model to learn two distinct representations. Nonetheless, the utility of RC depends heavily on the specific biological context. For instance, genomic elements such as splice sites are strictly defined by a single DNA strand, making RC inclusion potentially suboptimal or misleading.
>
> Regarding computational overhead, RC is introduced only during inference by averaging the embedding representations of a strand and its RC prior to decoding. Thus, the additional computational cost is minimal, primarily involving an extra forward pass through the model for the RC strand.
>
> To quantitatively assess the impact of incorporating RC, we conducted ablation experiments using the NT benchmark. We fine-tuned models under consistent experimental conditions (learning rate 1e-3, batch size 256), with and without RC during prediction. The results clearly indicate that models utilizing RC generally outperform those without RC across most tasks. Notably, the exception to this pattern was observed in splice site prediction tasks, where RC inclusion led to inferior performance, consistent with the biological reality that splice sites are inherently strand-specific.
>
> Based on your insightful feedback, we will clarify this discussion further in Section 3.2 and include the ablation experiment results explicitly in the final manuscript. We sincerely appreciate your constructive comments, which have helped strengthen our paper.
>
> |  | w/ attn; w/o rc | w/ attn; w/ rc | w/o attn; w/o rc | w/o attn; w/ rc |
> | --- | --- | --- | --- | --- |
> | H3 | 0.789±0.028 | **0.828±0.020** | 0.795±0.026 | **0.830±0.015** |
> | H3K14ac | 0.689±0.029 | **0.729±0.022** | 0.662±0.015 | **0.700±0.015** |
> | H3K36me3 | 0.661±0.021 | **0.701±0.022** | 0.658±0.016 | **0.688±0.012** |
> | H3K4me1 | 0.574±0.025 | **0.609±0.022** | 0.555±0.030 | **0.605±0.028** |
> | H3K4me2 | 0.546±0.026 | **0.588±0.020** | 0.532±0.020 | **0.581±0.024** |
> | H3K4me3 | 0.640±0.013 | **0.681±0.016** | 0.625±0.015 | **0.675±0.014** |
> | H3K79me3 | 0.723±0.025 | **0.747±0.013** | 0.710±0.020 | **0.743±0.009** |
> | H3K9ac | 0.638±0.023 | **0.673±0.014** | 0.631±0.016 | **0.658±0.020** |
> | H4 | 0.781±0.020 | **0.810±0.022** | 0.775±0.019 | **0.813±0.011** |
> | H4ac | 0.653±0.023 | **0.696±0.019** | 0.629±0.017 | **0.684±0.020** |
> | enhancers | 0.382±0.035 | **0.396±0.033** | 0.379±0.041 | **0.397±0.065** |
> | EnhancersTypes | 0.475±0.053 | **0.488±0.066** | 0.490±0.046 | **0.492±0.096** |
> | PromoterAll | 0.964±0.003 | **0.969±0.002** | 0.962±0.003 | **0.970±0.002** |
> | PromoterNoTata | 0.961±0.004 | **0.969±0.004** | 0.961±0.004 | **0.970±0.005** |
> | PromoterTata | 0.946±0.012 | **0.954±0.010** | 0.947±0.010 | **0.953±0.019** |
> | SpliceSitesAll | **0.961±0.003** | 0.948±0.008 | **0.946±0.007** | 0.922±0.019 |
> | SpliceSitesAcceptors | **0.928±0.010** | 0.906±0.016 | **0.932±0.014** | 0.904±0.008 |
> | SpliceSitesDonors | **0.915±0.008** | 0.893±0.006 | **0.921±0.009** | 0.874±0.011 |

---

> > ### Comment · Reviewer_tdwT · 2025-08-02
> >
> > Thank you for your thorough and thoughtful rebuttal. You have comprehensively addressed my concerns. Your explanation regarding the training speed is now clear and convincing; the architectural details clarifying the necessity of dense attention in the masked model baseline, versus the sparse attention in Janus modeling, fully resolve this point.
> > Furthermore, I was very impressed with the new ablation studies. The comparison between the Janus and CLM objectives provides strong, direct evidence for the superiority of your pre-training paradigm, significantly strengthening your claims. Similarly, the ablation on the reverse complement strategy adds a valuable and nuanced data-driven view of its utility. The new experimental results have resolved my primary reservations and have substantially strengthened the paper.

---

> > > ### Author Response · Authors · 2025-08-03
> > >
> > > Dear Reviewer tdwT,
> > >
> > > Thank you for your positive feedback and valuable suggestions. We are grateful for your time and expertise, which have significantly strengthened our paper. Please let us know if there are any additional points we could discuss.
> > >
> > > Many thanks! Have a nice day!
> > >
> > > Best wishes

---

### Official Review · Reviewer_yT7i · 2025-07-03

**Clarity:** 4
**Significance:** 3
**Originality:** 4
**Rating:** 5
**Confidence:** 4

**Summary:**

The paper proposes a new genomic foundation model (GFM), called JanusDNA, that seeks to improve upon existing GFMs by incorporating the bidirectional nature of DNA in the model architecture, and improves the training efficiency via Janus modelling.  In short, each sequence is processed by two independent stacks of Mamba feed-forward network and Mixture-of-Experts (MOE) layers, in the forward and reverse direction before being combined through a bidirectional fusion mask. The training objective seeks to predict every token, as opposed to just the masked tokens in Masked Language Modeling (MLM), which improves the training efficiency of JanusDNA. Evaluation on multiple genomic benchmarks demonstrates that JanusDNA supersedes existing models on a variety of tasks.

**Questions:**

**Questions:**

1.	Why is Janus pre-training faster than masked models (per 1000 steps)? On first looks, there appear to be more operations per training step in Janus modeling.
2.	Lines 200-204 makes a distinction between masked models and the Janus model. What is the difference between the T-th token being masked for masked models and it being omitted for the Janus model? Is this distinction meaningful?
3.	How is the JanusDNA model fine-tuned (line 317)?

**Suggestions:**

1.	SSM is first referred to in line 61 but the full form of this term is not stated until later. I recommend defining the acronym here.
2.	When the hybrid MoE is introduced in Section 3.2, it is not evident how this helps JanusDNA deal with longer sequences.

**Ethical Concerns:**

["NO or VERY MINOR ethics concerns only"]

**Final Justification:**

The additional clarifications and results have significantly strengthened the paper. I have raised my score accordingly to reflect this.

**Limitations:**

Yes

**Quality:**

3

**Strengths And Weaknesses:**

The core idea presented in the paper is sound and draws its motivation from biological domain knowledge. In particular, JanusDNA accounts for (a) the reverse complementarity of the DNA double-strand structure and (b) the bidirectional nature of context in DNA sequences.

The paper is also largely clear in its presentation of its ideas with a natural flow and good use of visualizations to aid the reader In understanding the various components of the proposed model. The use of formal mathematical notations also helps with the clarity of the presented ideas.

I would also like to laud the authors on including error bounds for the results in Tables 1 and 2.

While I do not believe there are significant weaknesses in this work, there are several questions that ought to be answered. Most of these have been delineated In a subsequent section but there are a couple of questions that I would like to raise here.

1.	In Eq, 6, how did the authors arrive at the current formulation of the cross-attention in the bidirectional fusion mask? The text refers to the need “to maintain the integrity of bidirectional prediction without future peeking” but it is not clear to me what this means.
2.	When discussing the training efficiency towards the end of Section 3.1, It is unclear what the metric for computational complexity is. There is reference to time taken per 1000 steps in the text, but Figure 3 plots the number of training steps. A well-established measure of complexity, clock time or FLOPs, would be more convincing here.

---

> ### Author Rebuttal · Authors · 2025-07-30
>
> Dear Reviewer yT7i,
>
> We sincerely appreciate your detailed review and recognition of our work. Below, we address your thoughtful and constructive suggestions.
>
> **Q1: Detailed explanation of Eq. 6**
>
> Thank you very much for highlighting this point; we are happy to clarify.
>
> First, the unidirectional representation follows an autoregressive structure, where, for example, the representation at position 0 predicts the token at position 1, the representation at position 1 predicts the token at position 2, and so forth.
>
> Thus, it is understandable that the masks in the self-attention regions (top-left and bottom-right in Figure 1(E)) allow attention to themselves, as reflected by the first and second lines in Eq. 6.
>
> Second, the cross-attention parts (top-right and bottom-left) indicate fusion of the two unidirectional representations. Similar to the forward autoregressive representation, the reverse representation uses the representation at position T to predict token T-1, position T-1 predicts token T-2, etc. Therefore, to predict a middle token (excluding tokens at positions 0 and T) using these bidirectional features, we account for an offset of "+2". For instance, when predicting token 5 (with indexing starting from 0), the forward representation at position 4 and the backward representation at position 6 are utilized. However, if here is "+1", for instance, when predicting token 5 (with indexing starting from 0), the forward representation at position 4 and the backward representation at position 5 are utilized, but the backward representation at position 5 includes the original information of token 5, causing information leakage, and training loss will decrease to 0 quickly correspondingly.
>
> We arrived at the formulation illustrated in Figure 1(E), which we have now improved to clearly indicate that the diagonal in the top-left and bottom-right region should also be unmasked. We also removed the confusing phrase “future peeking” in the manuscript and now clarify that we want to prevent information leakage. Thank you again for bringing this to our attention.
>
> **Q2: Metrics of computational complexity (refine figure)**
>
> Thank you for suggesting a more common complexity metric. We now additionally present the training efficiency based on the clock time required for model training along with corresponding performance on the last-token prediction task.
>
> Listing these details in a table may not be intuitive; therefore, in camera ready version of the paper, we update Figure 3 by plotting accuracy against training time to clarify this comparison. Thank you for this valuable recommendation.
>
> | Accuracy on Last Token Prediction |  |  |  |  |  |  |  |  |  |  |  |
> | --- | --- | --- | --- | --- | --- | --- | --- | --- | --- | --- | --- |
> |  | mask-32dim |  | janus-32dim |  | mask-64dim |  | janus-64dim |  | mask-128dim |  | janus-128dim |
> | Training time(mins) | 0.056 M | Training time(mins) | 0.056 M | Training time(mins) | 0.207 M | Training time(mins) | 0.207 M | Training time(mins) | 0.795 M | Training time(mins) | 0.795 M |
> | 60 | 0.404929 | 25.50 | 0.427821 | 60.5 | 0.4102496 | 25.50 | 0.439397 | 60 | 0.425551 | 25.50 | 0.444796 |
> | 120 | 0.434058 | 51.00 | 0.444575 | 121 | 0.4411103 | 51.00 | 0.451205 | 120 | 0.447582 | 51.00 | 0.459588 |
> | 180 | 0.441789 | 76.50 | 0.449629 | 181.5 | 0.4478824 | 76.50 | 0.461554 | 180 | 0.450395 | 76.50 | 0.471902 |
> | 240 | 0.443333 | 102.00 | 0.450901 | 242 | 0.4508481 | 102.00 | 0.466316 | 240 | 0.453748 | 102.00 | 0.473491 |
> | 300 | 0.444255 | 127.50 | 0.452616 | 302.5 | 0.4521624 | 127.50 | 0.470291 | 300 | 0.456034 | 127.50 | 0.479219 |
> | 360 | 0.447273 | 153.00 | 0.454893 | 363 | 0.4541665 | 153.00 | 0.472556 | 360 | 0.459998 | 153.00 | 0.481693 |
> | 420 | 0.449191 | 178.50 | 0.457761 | 423.5 | 0.4563688 | 178.50 | 0.474091 | 420 | 0.465365 | 178.50 | 0.484446 |
> | 480 | 0.450238 | 204.00 | 0.459449 | 484 | 0.4587279 | 204.00 | 0.477486 | 480 | 0.469885 | 204.00 | 0.48797 |
> | 540 | 0.451589 | 229.50 | 0.460683 | 544.5 | 0.4609852 | 229.50 | 0.478885 | 540 | 0.473288 | 229.50 | 0.490162 |
> | 600 | 0.452361 | 255.00 | 0.461319 | 605 | 0.461945 | 255.00 | 0.47961 | 600 | 0.474498 | 255.00 | 0.491077 |
>
> **Q3: Why is Janus pre-training faster than masked models (per 1000 steps)? On first looks, there appear to be more operations per training step in Janus modeling.**
>
> For this ablation experiment, to keep the comparison as fair as possible, we used the same architectures for both the Janus and the masked language model, except masks for the final attention layer.
>
> The Janus model consists of one layer with two parallel unidirectional encoders followed by a FlexAttention layer that incorporates a specially designed sparse attention mask for Janus-type feature fusion. As depicted in Figure 1(E) (which we revised to make the masking and prediction indices more intuitive to understand), the two unidirectional features are concatenated into a tensor of shape (batch, sequence_length × 2, dim).
>
> Compared to the Janus model, the architecture of the masked model is similar, where there is one layer of two separated parallel unidirectional encoders (same as Janus model) followed by a FlexAttention, but with no attention mask. This is because for masked training, partial tokens have been masked in the dataset loading process, causing the full attention requirement for bidirectional feature fusion. Meanwhile, to keep the comparison as fair as possible, the two unidirectional features from the layer of unidirectional encoders will also be concatenated as a tensor with the shape of (batch, sequence_length * 2, dim).
>
> Therefore, the speed difference primarily arises from the attention stage, where the Janus model employs sparse attention, while the masked model uses dense, full attention. Additionally, the masking pre-processing for the masked model incurs some computational overhead. Consequently, the Janus model achieves faster performance than the masked model when the architectures are identical.
>
> Importantly, an alternative implementation of the masked model would concatenate features along the feature axis rather than the sequence axis, resulting in final attention over a tensor with shape (batch, sequence_length, dim * 2). This approach would make the computational costs of Janus and masked pretraining nearly equivalent. However, we opted against this implementation in our ablation study because concatenating along the feature axis would alter the masked model's capacity, compromising the comparability between the two approaches. We have now clarified this point in the description of the ablation experiment.
>
> **Q4: Lines 200-204 makes a distinction between masked models and the Janus model. What is the difference between the T-th token being masked for masked models and it being omitted for the Janus model? Is this distinction meaningful?**
>
> Thank you for closely examining this point. There is indeed no distinction; all models receive identical inputs, where the last token is masked for prediction. We will revise this description in future versions to eliminate confusion. We appreciate your careful observation.
>
> **Q5: How is the JanusDNA model fine-tuned (line 317)?**
>
> For the DNALONGBENCH tasks, all models, including Caduceus and JanusDNA, were fine-tuned entirely (full model fine-tuning) using a learning rate of 4e-4, a batch size of 8, and one batch per GPU over three epochs. We specified these hyper-parameters explicitly in Appendix D.2, lines 579–583, and will ensure this is clearly stated in future revisions.
>
> Thank you for pointing out this ambiguity.
>
> **S1: SSM is first referred to in line 61 but the full form of this term is not stated until later. I recommend defining the acronym here.**
>
> Thank you for identifying this oversight. We will define the acronym "SSM" upon its first mention in line 61 in the camera ready version of the manuscript.
>
> **S2: When the hybrid MoE is introduced in Section 3.2, it is not evident how this helps JanusDNA deal with longer sequences.**
>
> We appreciate your insightful comment. Our intention was not to claim explicitly that MoE handles longer sequences more effectively. Rather, MoE models achieve superior performance compared to dense models with an equivalent number of inference parameters. This can be observed in Figure 4, where MoE models consistently demonstrate improved perplexity compared to dense models, across both shorter (1024) and longer (131072) sequence lengths.
>
> Meanwhile, for many downstream tasks of DNA language models, we need to run inference on the entire genome (3 billion nucleotide pairs), potentially for many individuals. Therefore inference performance is crucial, requiring better  performance while less running time, and MoE helps with that. We will clarify this point in the discussion in the camera ready version of the manuscript.
>
> Thank you once again for your valuable feedback. Your suggestions and comments will significantly enhance the quality of our manuscript, and we are committed to making the necessary improvements to meet the high standards of the NeurIPS community.

---

> > ### Comment · Reviewer_yT7i · 2025-08-05
> > **Convincing rebuttal**
> >
> > I thank the authors for addressing the detailed pointwise response to my concerns and those of the other reviewers. I believe the additional clarifications and results have strengthened the paper. I have raised my score accordingly to reflect this.

---

> ### Author Response · Authors · 2025-08-06
>
> Dear Reviewer yT7i,
>
> Thank you for your thoughtful feedback and for recognizing the improvements. We greatly appreciate your detailed comments, which have significantly strengthened our paper, and your raised score. Have a nice day!
>
> Best wishes

---

### Official Review · Reviewer_hQUx · 2025-07-06

**Clarity:** 3
**Significance:** 3
**Originality:** 4
**Rating:** 4
**Confidence:** 4

**Summary:**

JanusDNA introduces the first **bidirectional DNA foundation model** that overcomes key limitations in genomic sequence modeling: (1) inability to capture long-range dependencies (>10k bp) at single-nucleotide resolution, (2) unidirectional bias in existing models, and (3) low training efficiency of masked language modeling (MLM). Its core innovations include:

- **Hybrid Mamba-Attention-MoE architecture** for efficient million-bp-scale modeling.
- **Janus Modeling**, a novel pre-training paradigm combining autoregressive training efficiency with bidirectional comprehension.
- **Reverse-complement integration** to handle bidirectional genomic motifs.
Evaluated on 35 genomic tasks, it achieves SOTA on 12/18 Nucleotide Transformer tasks and 8/9 eQTL tasks despite using **7.66M activated parameters** (vs. 250× larger baselines).

**Questions:**

According to Figure 1B, the model processes both strands of the DNA double helix through left-to-right and right-to-left modeling. Given that DNA follows the complementary base-pairing principle (where one strand theoretically determines the sequence of its complement), simultaneous modeling of both strands appears redundant. Could you clarify the advantage and biological significance behind this design choice, or provide some ablation experiments?

**Ethical Concerns:**

["NO or VERY MINOR ethics concerns only"]

**Limitations:**

yes

**Quality:**

3

**Strengths And Weaknesses:**

### **Strengths & Contributions**

1. **Unprecedented sequence length & resolution**:
    - Processes **1 million bp** at single-nucleotide resolution on single 80GB GPU, demonstrating good efficiency.
2. **Architectural innovation**:
    - The Mamba-Attention-MoE design integrates local/global context modeling with sparse expert scaling, outperforming SSM/Transformer hybrids.
3. **Biological relevance**:
    - Explicit modeling of **reverse-complement strands** captures non-palindromic motifs, which is critical for gene regulation.
4. **Empirical rigor**:
    - Extensive ablation studies validate design choices, and SOTA results on eQTL prediction surpass specialist models like Enformer.

        ---


### **Limitations & Weaknesses**

1. **Narrow data scope**:
    - Trained *only* on the human reference genome (HG38) (Page 7, §4.1), limiting cross-species generalizability. And Lacks epigenetic data (e.g., chromatin states), noted as future work.
2. **Limited improvement in Genomic Benchmark**
    1. The model demonstrates suboptimal performance on the Genomic Benchmark, failing to surpass Caduceus baseline models in most tasks. While the authors claim this benchmark is 'saturated', recent work ConvNova [1] achieved significant performance improvements on these same tasks using CNN-based models with fewer parameters, outperforming multiple Mamba and Transformer-based models. The authors should provide comparative analysis against relevant models.

[1] Revisiting Convolution Architecture in the Realm of DNA Foundation Models, https://openreview.net/forum?id=B07dLVWLyD

---

> ### Author Rebuttal · Authors · 2025-07-30
>
> Dear Reviewer hQUx,
>
> We are very grateful for your detailed review and recognition of our work's contributions. We have carefully considered your comments and respond as follows:
>
> **W1: Narrow data scope.** We fully agree that the data scope of this work is limited to the human reference genome to achieve the fairest possible comparison with existing methods since the primary contribution of this work is the model architecture and pre-training paradigm itself as you acknowledged. We truly appreciate that. We fully agree that, for many DNA language model applications, diverse pre-training data is crucial and we are actively working on extending the data scope to genomes of additional species, more individual human genomes, and incorporating more modalities such as epigenetic information and different sequencing data types to make the JanusDNA architecture more generalizable and robust. Thank you very much for your suggestion.
>
> **W2: Limited improvement in Genomic Benchmark.** We greatly appreciate your recommendation of this reference paper. We have carefully reviewed it and acknowledge its powerful and novel approach and now include the ConvNova performance in our results tables.
>
> Additionally, we made further improvements to our architecture and reran all experiments. Specifically, we noticed that the model benefits from additional MLP layers after the final attention layer that fuses the forward and backward information as an example shows below.
>
> ```bash
> Sequential(
>       (0): Linear(in_features=32, out_features=64, bias=True)
>       (1): SiLU()
>       (2): Linear(in_features=64, out_features=32, bias=True)
>       (3): SiLU()
>     )
> ```
>
> This simple modification does not significantly change the model design or size, but most performance metrics improved considerably. We updated the manuscript with these new results for the cam-ready version.
>
> We attached the updated results for your reference, where `JanusDNA_mlp` is the latest model:
>
> ### Genomic Benchmark
>
> Following the settings from Caduceus, we fine-tuned the models with a batch size of 256, obtaining the new results shown in columns 5 and 6, surpassing ConvNova on 4 out of 8 tasks. Since ConvNova did not specify a batch size, we performed additional parameter searches across batch sizes of 128, 256, and 512. The results are shown in columns 7 and 8, surpassing ConvNova on 7 out of 8 tasks. (5-fold cross-validation).
>
> | 1 | 2 | 3 | 4 | 5 | 6 | 7 | 8 |
> | --- | --- | --- | --- | --- | --- | --- | --- |
> | batch |  | 256 | 256 | 256 | 256 | best among 128, 256, 512 | best among 128, 256, 512 |
> |  | ConvNova | JanusDNA | JanusDNA | JanusDNA_mlp | JanusDNA_mlp | JanusDNA_mlp | JanusDNA_mlp |
> |  |  | w/ Mid-Attn | w/o Mid-Attn | w/ Mid-Attn | w/o Mid-Attn | w/ Mid-Attn | w/o Mid-Attn |
> | Size | 386K | 422k | 427k | 426k | 431k | 426k | 431k |
> | MouseEnhancers | 0.784±0.009 | 0.752±0.041 | 0.750±0.031 | 0.770±0.048 | 0.769±0.029 | 0.798±0.029 | **0.802±0.050** |
> | HumanEnhancersCohn | 0.743±0.005 | 0.745±0.004 | 0.740±0.002 | 0.741±0.005 | 0.742±0.006 | **0.743±0.004** | **0.743±0.003** |
> | HumanEnhancerEnsembl | 0.900±0.004 | 0.899±0.003 | 0.900±0.005 | 0.897±0.004 | 0.899±0.004 | 0.901±0.003 | **0.904±0.001** |
> | Coding vs. Intergenomic | **0.943±0.001** | 0.913±0.001 | 0.914±0.002 | 0.912±0.003 | 0.911±0.001 | 0.912±0.003 | 0.913±0.002 |
> | Human vs. Worm | 0.967±0.002 | 0.972±0.001 | 0.971±0.001 | 0.971±0.001 | 0.971±0.001 | 0.971±0.001 | **0.971±0.001** |
> | HumanRegulatory | 0.873±0.002 | 0.877±0.002 | 0.870±0.006 | 0.877±0.005 | 0.868±0.008 | **0.877±0.003** | 0.869±0.002 |
> | HumanNonTATAPromoters | 0.951±0.003 | 0.949±0.006 | 0.945±0.006 | 0.957±0.004 | 0.954±0.010 | **0.957±0.004** | 0.954±0.008 |
> | HumanOCR_Ensembl | 0.793±0.004 | 0.825±0.002 | 0.825±0.003 | 0.822±0.003 | 0.824±0.001 | 0.823±0.003 | **0.824±0.003** |
>
> ### Nucleotide Transformer (NT)
>
> Performance on NT also improved significantly, as shown in columns 6 and 7. Columns 4 and 5 represent performance without the two additional simple MLP layers. (Columns 8 and 9 are for addressing Q1 to save space about ablations on reverse complement design.) (10-fold cross-validation)
>
> | 1 | 2 | 3 | 4 | 5 | 6 | 7 | 8 | 9 |
> | --- | --- | --- | --- | --- | --- | --- | --- | --- |
> |  | Caduceus-PH | Convnova | JanusDNA | JanusDNA | JanusDNA_mlp | JanusDNA_mlp | JanusDNA_mlp | JanusDNA_mlp |
> |  |  |  | w/ midattn | w/o midattn | w/ midattn; w/ rc | w/o midattn; w/ rc | w/ attn; w/o rc; 256, 1e-3 | w/ attn; w/ rc; 256, 1e-3 |
> | size(M) | 1.9 | 1.7 | 1.980 | 1.988 | 2.001 | 2.009 | 2.001  | 2.001 |
> | H3 | 0.815±0.048 | 0.812±0.017 | 0.821±0.021 | 0.824±0.012 | **0.835±0.009** | 0.831±0.023 | 0.789±0.028 | **0.828±0.020** |
> | H3K14ac | 0.631±0.026 | 0.644±0.009 | 0.665 ± 0.034 | 0.685±0.016 | **0.729±0.022** | 0.718±0.026 | 0.689±0.029 | **0.729±0.022** |
> | H3K36me3 | 0.601±0.129 | 0.661±0.019 | 0.658 ± 0.024 | 0.670±0.012 | **0.702±0.015** | 0.699±0.025 | 0.661±0.021 | **0.701±0.022** |
> | H3K4me1 | 0 .523±0.039 | 0.554±0.023 | 0.563 ± 0.041 | 0.571±0.018 | 0.615±0.035 | **0.616±0.018** | 0.574±0.025 | **0.609±0.022** |
> | H3K4me2 | 0.487±0.170 | 0.485±0.032 | 0.509 ± 0.056 | 0.548±0.022 | **0.589±0.023** | 0.586±0.019 | 0.546±0.026 | **0.588±0.020** |
> | H3K4me3 | 0.544 ±0.045 | 0.566±0.027 | 0.605 ± 0.030 | 0.629±0.022 | **0.688±0.026** | 0.675±0.014 | 0.640±0.013 | **0.681±0.016** |
> | H3K79me3 | 0.697±0.077 | 0.700±0.007 | 0.716 ± 0.017 | 0.727±0.023  | **0.747±0.013** | 0.743±0.009 | 0.723±0.025 | **0.747±0.013** |
> | H3K9ac | 0.622±0.030 | 0.658±0.011 | 0.641 ± 0.024 | 0.639±0.019 | **0.673±0.014** | 0.661±0.027 | 0.638±0.023 | **0.673±0.014** |
> | H4 | 0.811±0.022  | 0.808±0.008 | 0.809 ± 0.021 | **0.816±0.008** | 0.812±0.011 | 0.813±0.013 | 0.781±0.020 | **0.810±0.022** |
> | H4ac | 0.621±0.054 | 0.636±0.011 | 0.637±0.060 | 0.653±0.034 | 0.698±0.013 | **0.705±0.023** | 0.653±0.023 | **0.696±0.019** |
> | enhancers | 0 .546 ±0.073 | **0.586±0.038** | 0.564 ± 0.022 | 0.535±0.036 | 0.559±0.042 | 0.542±0.044 | 0.382±0.035 | **0.396±0.033** |
> | EnhancersTypes | 0.439±0.054 | 0.500±0.018 | 0.462±0.049 | 0.470±0.025 | 0.503±0.038 | 0.492±0.096 | 0.475±0.053 | **0.488±0.066** |
> | PromoterAll | 0.970±0.004 | 0.967±0.001 | 0.969±0.002 | **0.971±0.002** | 0.970±0.002 | 0.970±0.003 | 0.964±0.003 | **0.969±0.002** |
> | PromoterNoTata | 0.969±0.011 | 0.968±0.003 | 0.971±0.003 | **0.971±0.002** | 0.971±0.004 | 0.971±0.003 | 0.961±0.004 | **0.969±0.004** |
> | PromoterTata | 0.953±0.016 | **0.969±0.003** | 0.956±0.010 | 0.958±0.008 | 0.958±0.007 | 0.960±0.008 | 0.946±0.012 | **0.954±0.010** |
> | SpliceSitesAll | 0.940±0.027 | 0.965±0.004 | 0.963±0.022 | 0.960±0.009 | **0.967±0.005** | 0.943±0.020 | **0.961±0.003** | 0.948±0.008 |
> | SpliceSitesAcceptors | 0.937±0.033 | **0.971±0.003** | 0.949±0.020 | 0.939±0.022 | 0.957±0.012 | 0.961±0.009 | **0.928±0.010** | 0.906±0.016 |
> | SpliceSitesDonors | 0.948±0.025 | **0.965±0.003** | 0.947±0.015 | 0.936±0.014 | 0.948±0.008 | 0.935±0.016 | **0.915±0.008** | 0.893±0.006 |
>
> ### DNALONGBENCH
>
> Performance also improved significantly after adding the two simple MLP layers.
>
> |  | Caduceus-PH | JanusDNA; w/o midattn | JanusDNA_mlp; w/o midattn|
> | --- | --- | --- | --- |
> | size | 7.7M | 7.662 M | 7.745 M |
> | AT | 0.690 | 0.802 | **0.851** |
> | AS | 0.759 | 0.740 | **0.768** |
> | CCF | 0.689 | 0.770 | **0.801** |
> | MS | 0.789 | 0.803 | **0.864** |
> | NT | 0.841 | 0.877 | **0.913** |
> | SNSES | 0.812 | 0.874 | **0.903** |
> | SSELL | 0.691 | 0.706 | **0.845** |
> | Thyroid | 0.703 | 0.752 | **0.792** |
> | WB | 0.768 | 0.794 | **0.821** |
>
> **Q1: Value of reverse complement design and ablation?**
>
> Thank you very much for raising this important point. We fully agree that clarifying the value of the reverse complement (RC) design is crucial.
>
> As you correctly pointed out, DNA follows the complementary base-pairing principle, meaning each DNA strand has a reverse complement strand with equivalent genetic information. However, despite this theoretical equivalence, many biologically important motifs (e.g., transcription factor binding sites) are non-palindromic. Recognizing both the forward and RC versions of such motifs (for instance, motifs like GATA and TATC) is challenging, as it effectively requires the model to learn two distinct representations. Therefore, explicitly integrating RC information allows the model to more robustly and comprehensively capture DNA sequence patterns.
>
> Nonetheless, the usefulness of RC information can depend heavily on the specific biological context. For instance, certain genomic elements such as splice sites are defined strictly by a single DNA strand (the template strand), making the inclusion of RC information potentially suboptimal or even misleading.
>
> To quantitatively assess the impact of incorporating RC information, we have conducted additional ablation experiments using the NT benchmark. We fine-tuned all models under consistent experimental conditions (learning rate of 1e-3, batch size of 256) across 10-fold cross-validation, with the only difference being with or without of RC information during prediction.
>
> The results (presented in the last two columns of the above Table on NT, following the main experiments) clearly indicate that models with RC information generally outperform those without RC across most tasks. Notably, the exception to this pattern was observed in splice site tasks, where RC inclusion led to inferior performance, reflecting the biological reality that splice sites are inherently strand-specific.
>
> Based on your insightful suggestion, we will clarify and expand our discussion on this topic in Section 3.2 (Reverse Complement) and include this new experiment in the final manuscript. We truly appreciate your constructive feedback, which has enabled us to further strengthen our work.

---

### Decision · Program_Chairs · 2025-09-17

**Decision:**

Accept (poster)

**Comment:**

All reviewers recommended acceptance, citing the paper’s technical soundness, biological relevance, and strong empirical results. Concerns about training efficiency and generalization were well addressed during the rebuttal phase. The additional ablation studies and expanded discussion on biological motivations further strengthened the submission.